# Inhibition of post-termination ribosome recycling at premature termination codons in UPF1 ATPase mutants

**Lucas D Serdar[1,2], DaJuan L Whiteside[1], Sarah L Nock[1], David McGrath[1], Kristian E Baker[1,2]***

[1]Department of Genetics and Genome Sciences, Case Western Reserve University School of Medicine, Cleveland, United States; [2]Department of Biochemistry, Case Western Reserve University School of Medicine, Cleveland, United States

**Abstract** Recognition and rapid degradation of mRNA harboring premature translation termination codons (PTCs) serves to protect cells from accumulating non-functional and potentially toxic truncated polypeptides. Targeting of PTC-containing transcripts is mediated by the nonsense-mediated mRNA decay (NMD) pathway and requires a conserved set of proteins including UPF1, an RNA helicase whose ATPase activity is essential for NMD. Previously, we identified a functional interaction between the NMD machinery and terminating ribosomes based on 3' RNA decay fragments that accrue in UPF1 ATPase mutants. Herein, we show that those decay intermediates originate downstream of the PTC and harbor 80S ribosomes that migrate into the mRNA 3' UTR independent of canonical translation. Accumulation of 3' RNA decay fragments is determined by both RNA sequence downstream of the PTC and the inactivating mutation within the active site of UPF1. Our data reveal a failure in post-termination ribosome recycling in UPF1 ATPase mutants.

***For correspondence:**
kristian.baker@case.edu

**Competing interests:** The authors declare that no competing interests exist.

## Introduction

Messenger RNA (mRNA) degradation provides a robust means to regulate gene expression and limit the quantity of protein produced from information transcribed from individual genes. It has long been recognized that the decay rates of mRNAs within a single cell can differ by orders of magnitude and that the stability of a transcript is tightly coupled to its translation (*Bicknell and Ricci, 2017*). What is only beginning to emerge is an appreciation for the complexity of the relationship between mRNA translation and decay, and how RNA primary sequence and structure interplay with the multitude of RNA binding proteins capable of influencing translation at one or more of the initiation, elongation or termination steps of protein synthesis (*Heck and Wilusz, 2018*; *Schoenberg and Maquat, 2012*). One well-documented example of translation-dependent modulation of mRNA stability is the accelerated degradation of transcripts harboring premature translation termination codons (PTCs) by the nonsense-mediated mRNA decay (NMD) pathway. The cell's ability to discern premature termination events and rapidly eliminate PTC-containing transcripts serves a vital quality control function by preventing the accumulation of carboxy-terminal truncated polypeptides that, depending on the point of truncation, likely lack function or, in some cases, acquire detrimental gain-of-function activity. Despite our appreciation of NMD and its broad role in modulating the levels of ~10% of cellular transcriptomes (*Celik et al., 2017*; *Mendell et al., 2004*; *Rehwinkel et al., 2005*; *Smith et al., 2014*), our mechanistic understanding of how translation termination is perceived as premature and the subsequent events that transpire to signal rapid decay of the mRNA remain unclear.

The machinery responsible for mediating NMD consists of three core and highly conserved proteins, UPF1, UPF2, and UPF3 (*Kervestin and Jacobson, 2012*). How the NMD machinery associates

with substrates and monitors translation termination has been the focus of significant debate and is envisioned by two prevailing models (*He and Jacobson, 2015*). The first posits that premature termination, which by default occurs in the context of a lengthened or *faux* 3' UTR, is inherently inefficient and results in a delay in ribosome-associated events at the PTC that is sufficient to cause subsequent recruitment and/or activation of UPF proteins on the translation machinery. The second model proposes that NMD components assemble indiscriminately on most or all transcripts, but are displaced from protein coding regions by elongating ribosomes so as to accumulate preferentially on transcripts in a 3' UTR length-dependent manner, where they are poised to interact with terminating ribosomes and elicit downstream events. This latter model is supported by both the observed enhancement of NMD by exon-junction complexes (of which UPF3 is a peripheral component) and genome-wide binding studies revealing UPF1 binding to both normal and NMD-sensitive mRNA and redistribution of the protein from sites predominantly within 3' UTRs to those along the entire transcript body upon inhibition of translation (*Hurt et al., 2013*; *Zünd et al., 2013*). Extended 3' UTRs derived from the abbreviated open reading frame of PTC-containing mRNA, therefore, serve as a preferential binding platform for UPF protein interaction and provide a rationale for how a premature translation termination event is preferentially targeted by this pathway.

Independent of the mode of UPF protein association with mRNA, the translation-dependent nature of NMD specifies that UPF protein binding is insufficient to elicit NMD and that a functional interaction between the NMD and translation machinery must occur before initiating degradation of the mRNA. Such an interface between the NMD and translational machineries is supported by biochemical data demonstrating the interaction of one or more UPF proteins with ribosomes, ribosomal proteins, or rRNA (*Min et al., 2013*; *Schuller et al., 2018*) and with eukaryotic release factors 1 and 2 (eRF1 and eRF3), proteins involved in stop codon recognition and nascent peptide hydrolysis during translation termination (*Ivanov et al., 2008*; *Kashima et al., 2006*; *Neu-Yilik et al., 2017*; *Singh et al., 2008*; *Wang et al., 2001*). Moreover, evidence for UPF1 protein involvement in stop codon readthrough in yeast (*Weng et al., 1996a*; *Weng et al., 1996b*) and translation termination efficiency in cell-free extracts (*Amrani et al., 2004*; *Ghosh et al., 2010*) provides functional data for NMD components modulating ribosome activity. Despite these observations, recent studies using purified components and reconstituted translation assays have failed to assign a role for UPF1 in influencing either the efficiency of termination or subsequent ribosome subunit recycling (*Neu-Yilik et al., 2017*; *Schuller et al., 2018*), leaving our understanding of this critical step in NMD incomplete.

UPF1 is a member of the SF1 family of RNA helicases and exhibits RNA binding and ATP hydrolysis activities, both of which are required for NMD. Mutation of conserved residues within the UPF1 ATP binding pocket that abrogate either nucleotide binding or hydrolysis leads to stabilization of NMD substrate mRNA (*Weng et al., 1996a*). Structural studies on both yeast and human UPF1 (*Chakrabarti et al., 2011*) have illuminated how ATP binding and hydrolysis invoke conformational changes to the protein that are thought to underlie the RNA unwinding and translocation activities observed for UPF1 in vitro (*Czaplinski et al., 1995*; *Fiorini et al., 2015*) and mRNA target discrimination and ribonucleoprotein (mRNP) remodeling in vivo (*Franks et al., 2010*; *Kurosaki et al., 2014*; *Lee et al., 2015*). Our previous work revealed that in yeast, failure of UPF1 to hydrolyze ATP leads to the accumulation of 3' RNA decay fragments from nonsense-containing mRNAs that arise due to an impediment in 5' → 3' degradation by exoribonuclease XRN1 (*Serdar et al., 2016*). The dependency on PTC position, translation of the mRNA, and for NMD cofactors UPF2 and UPF3 for fragment accumulation led us to propose that the block to XRN1 was a consequence of a stalled trimeric mRNP complex that forms between the ribosome, mRNA, and the NMD machinery, and that ATP hydrolysis by UPF1 is critical for efficient translation termination at a premature stop codon (*Serdar et al., 2016*). These data revealed an important functional interaction between the NMD and translation machinery and provided a direct role for ATP hydrolysis by UPF1 in events occurring during premature termination.

To gain mechanistic insight into how UPF1 impinges upon the translation machinery, we further characterized the 3' RNA fragments that accumulate in ATPase-deficient UPF1 mutants. Herein we provide evidence that the 5' terminus of decay intermediates originate downstream of the premature termination codon (PTC) and are distinct from fragments that amass when ribosomes are stalled at the PTC due to depletion of translation termination factor eRF1. Consistent with a defect that is distinct from stop codon recognition, we show that decay intermediates co-purify with 80S

ribosomes and that occupancy of the translation machinery on the mRNA 3' UTR is independent of canonical translation events. Further, we demonstrate that accumulation of decay fragments is dependent upon RNA sequence downstream of the PTC predicted to contact the mRNA binding channel of a ribosome residing on the 5' end of the fragment and that mutations within the active site of UPF1 alter the abundance and position of the 5' end of the RNA intermediates in an allele-specific manner. Our data reveal that a failure to hydrolyze ATP by UPF1 results in a defect in post-termination ribosome recycling at PTCs and migration of ribosomes into the mRNA 3' UTR, and provide novel evidence linking UPF1 catalytic activity with ribosome dynamics during premature termination.

## Results

### Decay fragments in ATPase-deficient UPF1 mutants derive downstream of the premature stop codon

We have shown previously that yeast cells expressing ATPase-deficient UPF1 (from the *UPF1 DE572AA* mutant allele) accumulate 3' RNA decay fragments arising from nonsense-containing mRNA whose size is dependent upon and coincident with the position of the PTC (*Serdar et al., 2016*). The requirement for mRNA translation in fragment formation and their co-sedimentation with 80S monosomes established an association between these RNA intermediates and ribosomes, and supported the conclusion that ATP hydrolysis by UPF1 was required for efficient translation termination at premature stop codons (*Serdar et al., 2016*). To better understand the nature of this functional interaction between the core NMD factor and premature terminating ribosomes, we employed primer extension analysis to identify the 5' terminus of the 3' RNA decay fragments. Characterization of products derived from *GFP* reporter mRNA harboring a PTC at codon position 125 (i.e. *GFP^PTC125* mRNA) on high-resolution polyacrylamide gels revealed cDNA corresponding to full-length mRNA in both wild type and UPF1 ATPase mutants (*Figure 1A*, lane 1 and 2, respectively; FL). Additional shorter cDNA products were detected for *UPF1 DE572AA* mutant cells where 3' RNA decay intermediates accumulate but not from wild-type cells. Strikingly, mapping of the termini of these shorter species revealed 5' ends corresponding to positions 17 and 20 nucleotides *downstream* of the UAA premature termination codon (*Figure 1A*, +17 and +20). Based on our earlier observation that 3' RNA decay fragments accrue as a result of a block in 5' → 3' exonucleolytic digestion catalyzed by XRN1 (*Serdar et al., 2016*), these data indicate that the impediment to complete degradation of the nonsense-containing mRNA is located downstream of the site of premature translation termination.

To confirm that the 5' termini of the decay fragments identified by primer extension are representative of RNA intermediates that accumulate in vivo, primer extension analysis was performed on RNA from cells depleted for eukaryotic release factor 1 (eRF1), a deficiency of which leads to inefficient translation termination due to impaired stop codon recognition and ribosome stalling with the stop codon positioned within the aminoacyl tRNA acceptor site (A site) (*Brown et al., 2015*). Chromosomally-encoded *SUP45* (encoding yeast eRF1) was placed under the control of a galactose-inducible promoter and expression inhibited by the growth of cells in media lacking galactose. Ten hours after inhibition of transcription, eRF1 levels were reduced to ~10% of steady-state (*Figure 1—figure supplement 1A*) and cells accumulated a 3' RNA intermediate from *GFP^PTC125* mRNA comparable to that observed in the *UPF1* ATPase mutant (*Figure 1—figure supplement 1B*). Analysis of primer extension products from these cells revealed full-length *GFP* cDNA and three additional products with 5' ends mapping 17, 47, and 77 nucleotides *upstream* of the PTC (*Figure 1A* lane 3; *GAL1-SUP45*). These data are consistent with co-translational decay products accumulating as a consequence of ribosome stalling at the PTC in the absence of eRF1 and protection of 17 nucleotides of RNA upstream of the A site codon (*Pelechano et al., 2015*). Moreover, the cDNA products at −47 and −77 reveal a periodicity of 30 nucleotides, indicative of a queuing of ribosomes upstream of the one arrested during termination at the PTC (*Figure 1B*). These data confirm cDNA 5' termini observed by primer extension analysis represent *bona fide* ends of 3' RNA decay fragments accumulating in vivo and highlight that the impediment to complete degradation of the nonsense-containing mRNA is distinct between cells expressing ATPase-deficient UPF1 and those impaired for translation termination due to depletion of eRF1. Further, these findings signify that the failure of

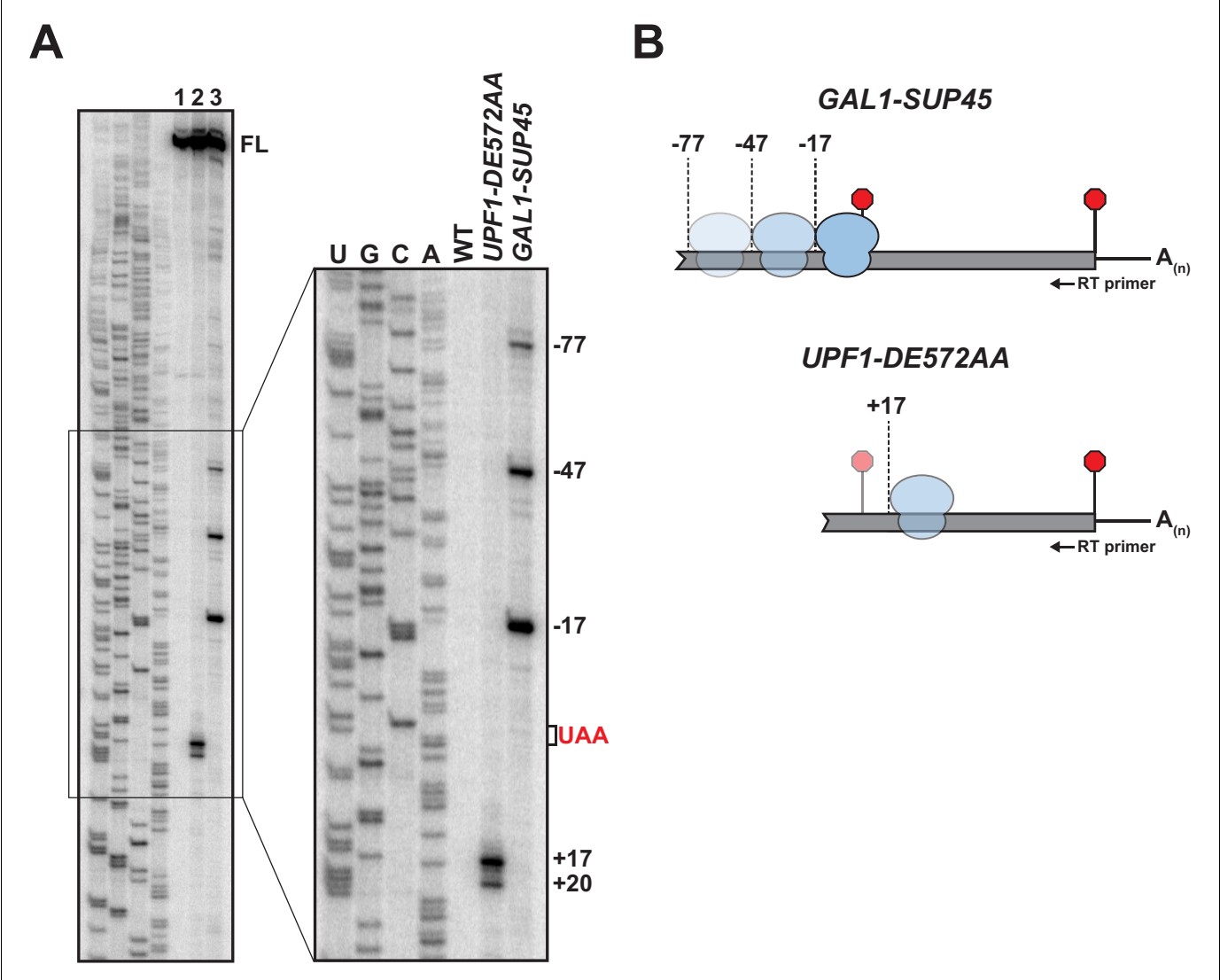

**Figure 1.** 5′ termini of RNA decay fragments in UPF1 ATPase derive downstream of the PTC. (**A**) Primer extension analysis of *GFP^PTC125* mRNA from cells expressing either wild type (WT) or ATPase-deficient UPF1 (*UPF1-DE572AA*), or depleted for translation termination factor eRF1 (*GAL1-SUP45*). Full-length cDNAs (FL) and products from RNA decay intermediates are indicated relative to the position of the premature stop codon (UAA, in red). (**B**) Schematic representation of the predicted sites of ribosome stalling in eRF1-depleted cells (top) and cells expressing ATPase-deficient UPF1 (bottom).

The online version of this article includes the following figure supplement(s) for figure 1:

**Figure supplement 1.** 5′ termini of RNA decay fragments in UPF1 ATPase are downstream of the PTC.

UPF1 to hydrolyze ATP does not preclude the function of release factors or cause stalling of ribosomes during translation termination at the PTC.

The observation that 3′ RNA decay fragments in UPF1 ATPase mutants have 5′ ends downstream of the PTC was unanticipated. To confirm that this observation is not specific to *GFP^PTC125* mRNA, primer extension analysis was performed on RNA from cells expressing two additional nonsense-containing mRNAs that we showed previously also accumulate 3′ decay intermediates in *UPF1 DE572AA* mutant cells (*Serdar et al., 2016*). Strikingly, cDNA products from *GFP* transcripts with a PTC at codon 67 or 135 also revealed decay products with 5′ ends that map downstream of the premature termination codon (*Figure 1—figure supplement 1C & D*), although RNA intermediates from these reporters display greater heterogeneity and the distance of their 5′ termini from the PTC are different from each other and from *GFP^PTC125* mRNA.

## 3' RNA decay intermediates are ribosome bound

We previously showed that 3' RNA decay fragments from PTC-containing mRNA in UPF1 ATPase mutants co-sediment with 80S monosomes in sucrose density gradients (i.e. polyribosome analysis), providing strong evidence that the intermediates were ribosome bound (*Serdar et al., 2016*). However, in light of our observation that the 5' ends of these decay products map downstream of the PTC and the protein-coding region for this transcript, we performed affinity purification of ribosomes to determine whether a physical association between the 3' decay fragments and the translational machinery could be demonstrated. Chromosomally-encoded small ribosomal protein gene *RPS13* was epitope-tagged at the protein's carboxy-terminus as previously described (*Min et al., 2013*) in cells harboring *UPF1 DE572AA* and the *GFP^PTC125* reporter. Immunoprecipitation of RPS13 from these cells recovered predominantly intact ribosomes as demonstrated by the presence of stoichiometric amounts of 25S and 18S ribosomal RNA in the recovered material (*Figure 2—figure supplement 1A*). Northern blot analysis for *GFP^PTC125* mRNA demonstrated co-precipitation of both full-length mRNA and the 3' RNA decay fragment, dependent upon the tagged ribosomal protein and mutant UPF1 (*Figure 2A*, compare lanes 3, 6, and 9), establishing that the decay intermediate is indeed ribosome bound. Consistent with this result and that the 3' RNA decay fragments associate with intact 80S ribosomes, immunoprecipitation of epitope-tagged large ribosomal protein RPL16a also co-purified the 3' degradation intermediate (*Figure 2—figure supplement 1B*).

To examine the 5' termini of ribosome-bound 3' RNA decay fragments, primer extension analysis was performed on RNA recovered by RPS13 immunopurification. Consistent with our observations using whole-cell RNA isolated from the UPF1 ATPase mutant, cDNA products from the precipitate also mapped 17 and 20 nucleotides downstream of the premature termination codon (*Figure 2B*, lane 6). These data corroborate our previous findings employing polyribosome analysis (*Serdar et al., 2016*) and demonstrate that the 3' RNA decay fragments which accumulate when UPF1 fails to hydrolyze ATP are ribosome bound, thereby establishing that these ribosomes reside on the RNA downstream of the premature stop codon within the mRNA 3' UTR (*Figure 1B*).

## Ribosome association downstream of the PTC does not occur via canonical translation events

To explain how the translation machinery associates with the 3' RNA decay intermediates in the UPF1 ATPase mutant, we sought to determine whether ribosomes undergo either read through past the PTC or engage in translation re-initiation after termination at the nonsense codon. To facilitate the detection of a protein product translated from the mRNA 3' UTR, nucleotides downstream of the PTC in *GFP^PTC125* were substituted with a sequence encoding an in-frame AUG codon and the 8 amino acid FLAG polypeptide (*Figure 3A*). The resulting *GFP^PTC125-FLAG* reporter was introduced into cells and expression of an internally FLAG-tagged protein of ~27 kDa was monitored by western blot. Notably, translational read-through products were not detected from either wild-type cells or mutants deleted for *UPF1* (i.e. *upf1Δ*), indicating efficient translation termination at the PTC in the presence of an active NMD pathway and in NMD-deficient cells lacking its core factor (*Figure 3B*, lanes 1 and 2). Moreover, in ATPase-deficient UPF1 mutants where ribosome-bound 3' RNA decay intermediates accumulate, translation read-through products were also absent (*Figure 3B*, lane 3). In contrast, a 27 kDa GFP-FLAG peptide accumulated to high levels in control cells depleted for eRF1 (*Figure 3B*, lane 4), indicative of inefficient termination at the PTC in the absence of this factor and continued translational elongation in-frame beyond the nonsense codon to the natural *GFP* stop codon. Importantly, the *GFP^PTC125-FLAG* reporter mRNA was confirmed to be efficiently translated in all four cell types as monitored by the accumulation of a truncated GFP polypeptide of 14 kDa (encoded by codons 1–125; *Figure 3B*, lanes 1–4).

To evaluate whether ribosomes associated with the RNA decay fragments are competent to re-initiate translation with the mRNA 3' UTR, AUG-FLAG sequences were introduced also into the +1 and +2 frames downstream of the PTC in the *GFP^PTC125* reporter (*Figure 3A*). Protein analysis for GFP-FLAG peptides initiated at the downstream AUG in the 0, +1, or +2 reading frame failed to detect expression of a ~ 13 kDa peptide in any of the cells tested (*Figure 3B*, lanes 1–12), despite efficient translation of the upstream *GFP* protein coding sequence (*Figure 3B*). These analyses of *GFP^PTC125-FLAG* reporters indicate that the ribosome-association of 3' decay fragments observed in

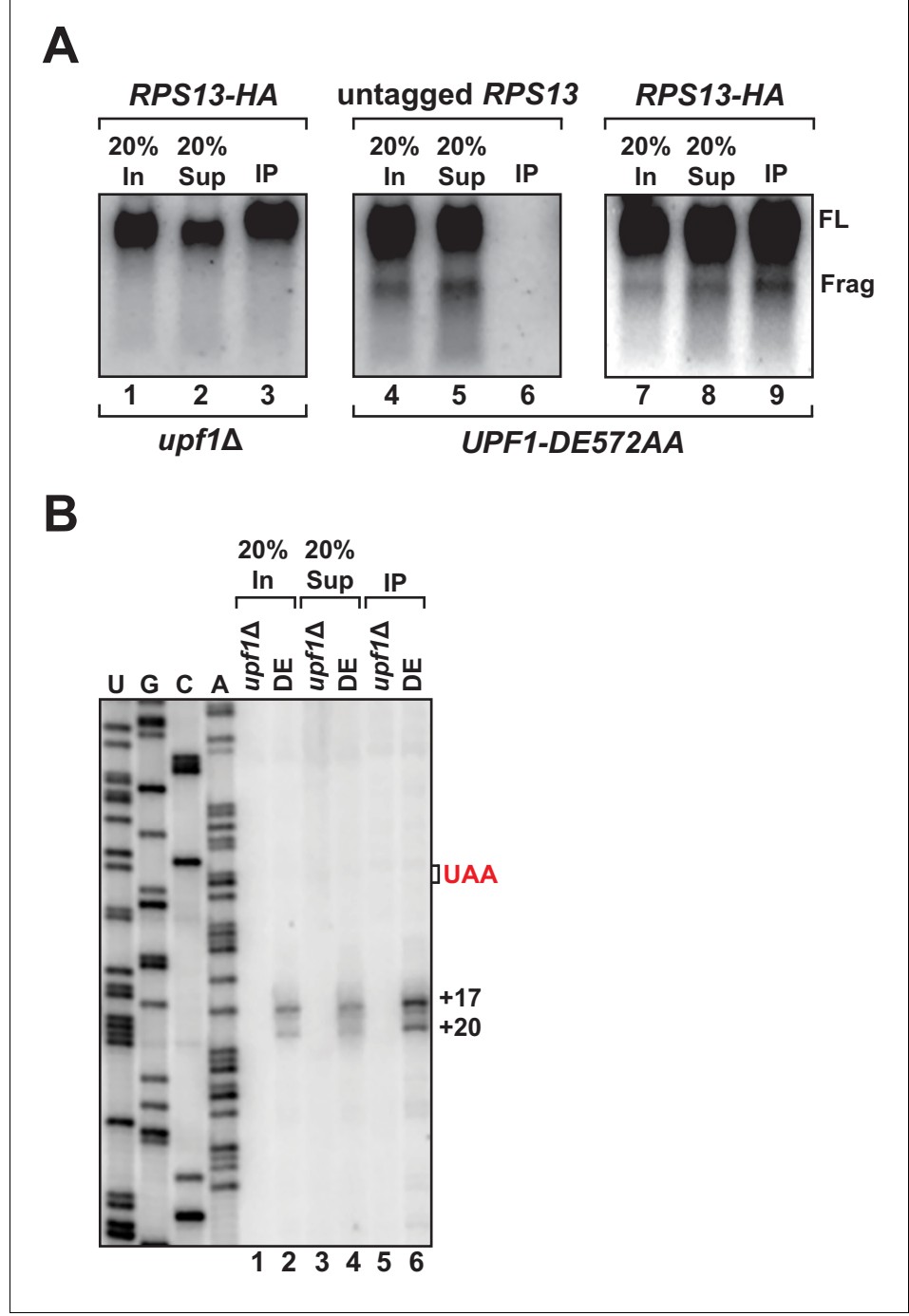

**Figure 2.** 3' RNA decay intermediates are ribosome bound. (**A**) Northern blot analysis of $GFP^{PTC125}$ mRNA co-immunopurified with untagged or epitope-tagged RPS13 (*RPS13-HA*) in cells deleted for *UPF1* (*upf1Δ*) or expressing the ATPase-deficient mutant, *UPF1-DE572AA*. Samples include input (In), supernatant (Sup) and immunopurified (IP) material; presence of 18S and 25S ribosomal RNA indicated. (**B**) Primer extension analysis of RNA samples from immunoprecipitation of RPS13-HA tagged ribosomes shown in (**A**). cDNA products from RNA decay intermediates are indicated relative to the position of the premature stop codon (UAA, in red).
The online version of this article includes the following figure supplement(s) for figure 2:

**Figure supplement 1.** 3' RNA decay intermediates are ribosome bound.

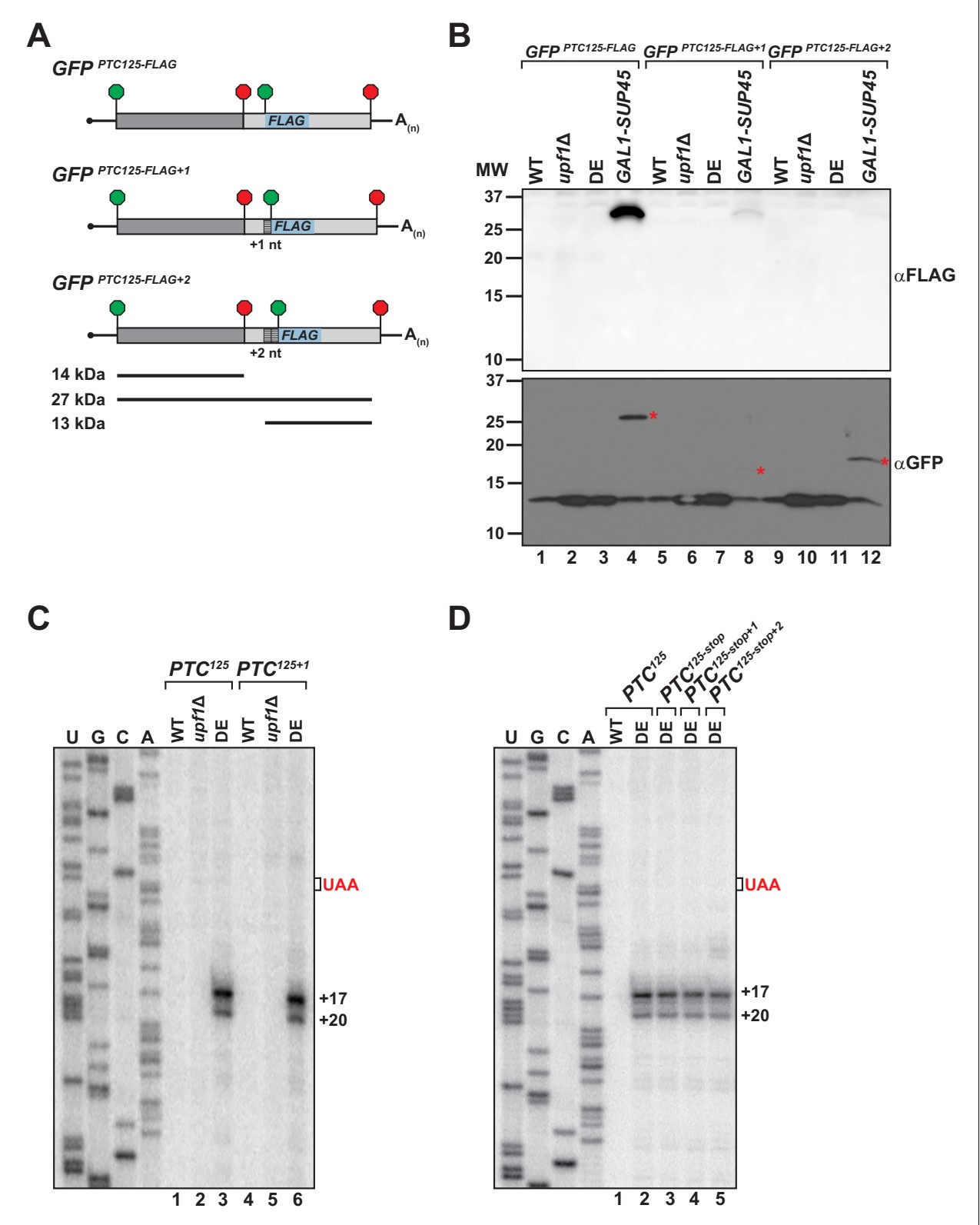

**Figure 3.** Translational read-through or reinitiation products are not detected in ATPase-deficient UPF1 mutants. (A) $GFP^{PTC125}$ reporters encoding an internal FLAG epitope downstream of the PTC in each translational reading frame. Predicted products for translation of $GFP$ mRNA to the PTC (14 kDa), read-through to the natural stop codon (27 kDa), or from reinitiation upstream of the FLAG sequence (13 kDa) are shown. (B) Western blot analysis for FLAG-tagged protein products (top) or GFP protein (bottom) from wild type (WT), UPF1 deletion cells ($upf1\Delta$), UPF1 ATPase mutants (DE)

*Figure 3 continued on next page*

*Figure 3 continued*

and eRF1-depleted cells (*GAL1-SUP45*) expressing the *GFP^PTC125-FLAG* reporters. Red asterisks indicate GFP polypeptides expected for translational read-through at the PTC for the three *GFP^PTC125-FLAG* reporters; size disparity reflect differences in translation termination of these products due to +1 and +2 shifts in the reading frames. (C) Primer extension analysis of *GFP^PTC125* and *GFP^PTC125+1* mRNA from WT, *upf1Δ*, and UPF1 ATPase mutants (DE). *GFP^PTC125+1* mRNA harbors a single-nucleotide insertion immediately downstream of the PTC. (D) Primer extension analysis of *GFP^PTC125* mRNA harboring a second stop codon in each translational reading frame beginning either 10, 11, or 12 nucleotides downstream of the PTC.
The online version of this article includes the following figure supplement(s) for figure 3:

**Figure supplement 1.** Alteration of translational reading frame downstream of the PTC does not alter primer extension products.

UPF1 ATPase mutants is not likely a consequence of translation read through and that 3′ UTR-associated ribosomes are unlikely competent to engage in translational re-initiation.

To assess whether reading frame downstream of the PTC influences the nature of the 3′ decay fragments, a single nucleotide was inserted immediately downstream of the nonsense codon of PTC-containing *GFP* reporters and the 5′ termini of the decay intermediates in *UPF1 DE572AA* mutants mapped by primer extension. Critically, cDNA products from *GFP^PTC125+1* mRNA (*Figure 3C*, lane 6) and *GFP^PTC135+1* mRNA (*Figure 3—figure supplement 1A*, lane 6) were identical to that observed for the parental reporters, indicating that canonical translation and ribosome translocation into the *GFP* 3′ UTR is also unlikely to account for how ribosomes associate with these 3′ decay fragments. Supporting this notion, introduction of a second premature termination codon in any of the three reading frames beginning either 10, 11, or 12 nucleotides downstream of PTC^125 also failed to alter the accumulation or nature of the 5′ termini of *GFP^PTC125* mRNA decay intermediates (*Figure 3D*, compare lanes 2 with 3–5).

## 3′ UTR sequence context determines the nature of 3′ RNA decay intermediates

Examination of primer extension products for various nonsense-containing reporter mRNAs revealed that while the 5′ termini of the decay intermediates that accumulate in UPF1 ATPase mutant cells were invariably downstream of the PTC, the number and distance from the translation termination site differed (*Figure 1A* and *Figure 1—figure supplement 1C and D*). Given that the 3′ decay fragments accumulate as a result of an impediment in 5′ → 3′ exonucleolytic decay (*Serdar et al., 2016*), we were interested in understanding the nature of this blockage and how its position is determined for a given mRNA. We observed that the insertion of a single-nucleotide downstream of the PTC did not shift the position of the 5′ ends of the decay intermediates formed from *GFP^PTC125+1* mRNA suggesting that the impediment is not determined by a fixed distance downstream of the PTC. To evaluate this further, three or six nucleotides were either inserted or deleted downstream of the PTC in the *GFP^PTC125* reporter and the termini of the decay intermediates mapped by primer extension. Notably, neither lengthening nor shortening the region directly downstream of the PTC by these lengths altered the position of the 5′ ends of the decay intermediates (*Figure 4A*), demonstrating that the site of the block to 5′ → 3′ decay remained unchanged and that the nucleotide distance from the PTC is not the sole factor in determining how XRN1 progression is impeded.

We next evaluated whether the 5′ terminus of the decay intermediates is dependent exclusively upon the RNA sequence downstream of the PTC. Two *GFP^PTC67* reporters were generated that harbor identical 5′ UTRs and protein-coding regions but with deletions within their 3′ UTRs that either maintained the sequence directly downstream of the PTC at codon 67 (i.e. *GFP^PTC67ΔB*) or juxtaposed sequences that would normally be present downstream of codon 125 (i.e. *GFP^PTC67ΔA*; *Figure 4B*, top). Primer extension products derived from these reporters in ATPase-deficient UPF1 cells revealed that the sequence downstream of the PTC indeed influences the nature and position of the 5′ termini of the decay intermediates. Notably, for *GFP^PTC67ΔB* mRNA, which contains a deletion internal in the 3′ UTR region distal to the PTC, the termini of the decay intermediates mirror that of the parental *GFP^PTC67* mRNA (compare *Figure 4B* left and *Figure 1—figure supplement 1C*). In contrast, deletion of a region immediately proximal to the PTC that introduce sequence originally downstream of *GFP* codon 125 leads to 3′ decay intermediates with termini identical to that of *GFP^PTC125* (compare *Figure 4B* right and *Figure 1A*). These data indicate that the blockage to 5′ →

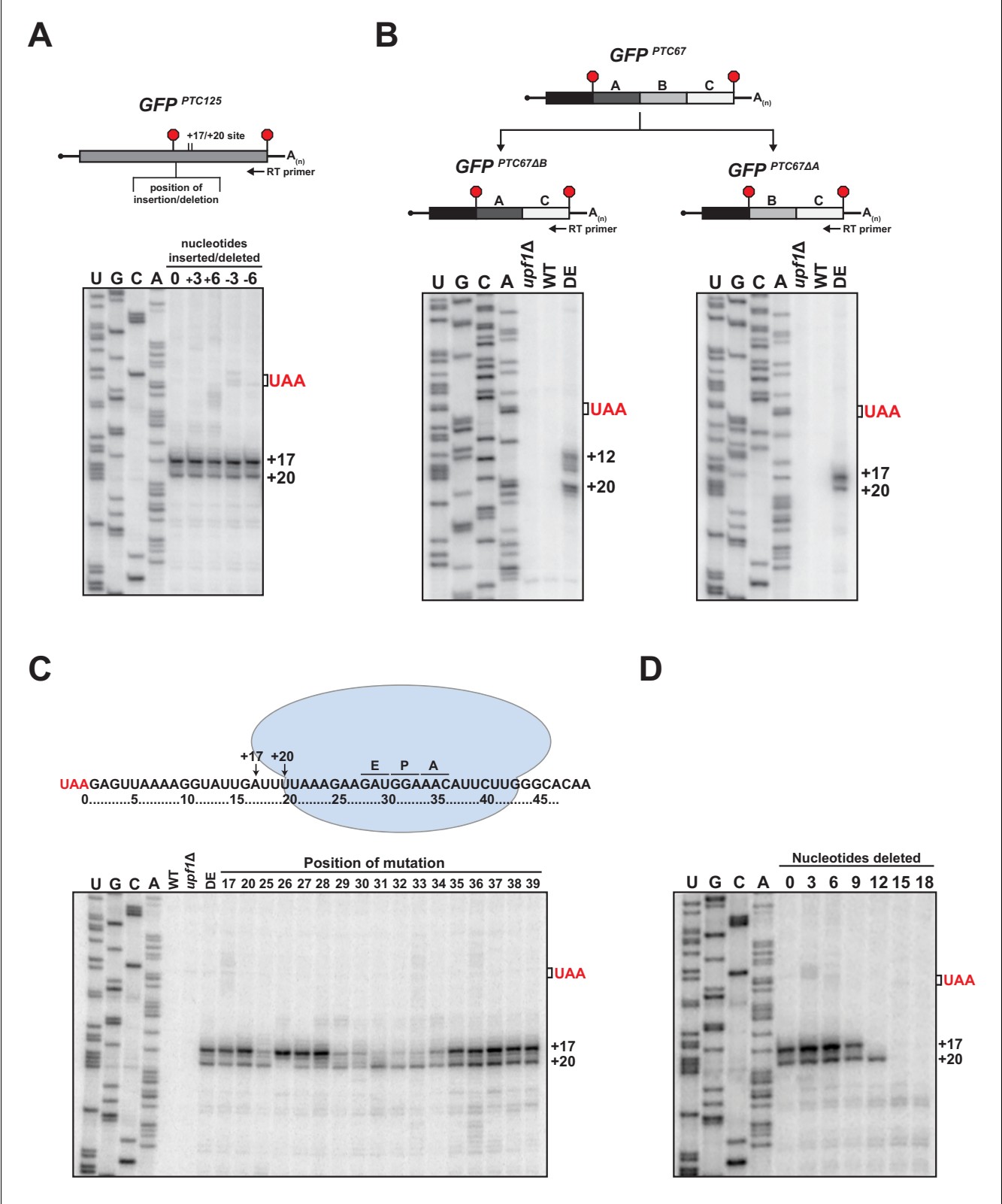

**Figure 4.** RNA sequence requirements for the accumulation of 3′ RNA decay fragments. (**A**) Schematic of nucleotide insertions or deletions immediately downstream of the PTC in *GFP^PTC125^* (top) and primer extension analysis of these reporter mRNAs from ATPase-deficient UPF1 mutants (bottom). (**B**) *GFP^PTC67^* reporters lacking ~170 nt within an internal 3′ UTR region (*GFP^PTC67ΔB^*) or just downstream of the PTC (*GFP^PTC67ΔA^*) and primer extension analysis of these RNAs from UPF1 deletion cells (*upf1Δ*) or cells expressing wild type (WT) or ATPase-deficient UPF1 (DE). (**C**) Schematic

*Figure 4 continued on next page*

Figure 4 continued

representation of *GFP^PTC125* reporter mRNA and stalled ribosome whose 30 nt footprint would protect RNA from +17 to +46; predicted positions for the ribosome A, P, and E sites are indicated (top). Primer extension analysis of *GFP^PTC125* reporters in wild type, UPF1 deletion cells (*upf1Δ*), or UPF1 ATPase mutants (DE) and various *GFP^PTC125* mRNAs harboring single-nucleotide inversions (at positions indicated) from ATPase-deficient UPF1 cells. (D) Primer extension analysis of *GFP^PTC125* mRNAs deleted for nucleotides immediately downstream of PTC in UPF1 ATPase mutants.

The online version of this article includes the following figure supplement(s) for figure 4:

**Figure supplement 1.** Accumulation of 3′ RNA decay intermediates in UPF1 ATPase mutants is unaffected by the nucleotide sequence of the PTC or penultimate sense codon.

3′ decay that gives rise to the 5′ ends of the decay intermediates is dependent specifically upon sequence context within the 3′ UTR proximal to the PTC.

To further dissect how *cis*-acting RNA sequence might influence the 3′ decay intermediates that accumulate in UPF1 ATPase-deficient cells, we introduced a number of point mutations into the *GFP^PTC125* reporter and evaluated how these changes impact the 5′ termini of decay intermediates that accrue from the corresponding mRNAs. We first evaluated the premature stop codon itself by mutating the ochre stop (UAA) in *GFP^PTC125* mRNA to either UAG (amber) or UGA (opal). Notably, primer extension products for all three of these reporters were identical (*Figure 4—figure supplement 1A*), signifying that accumulation of the 5′ end of the decay intermediates is independent of the identity of the premature stop codon. We further evaluated whether the nature of the codon just upstream of the PTC was important in 3′ decay intermediate accumulation, given reports that post-termination ribosomes can migrate along mRNA in vitro to codons cognate to the tRNA in their P site (*Skabkin et al., 2013*). Mutation of the codon penultimate to the PTC within *GFP^PTC125* from the AUC codon in wild-type *GFP* to four distinct codons also did not effect the accumulation or position of the 5′ termini of the decay intermediates found in ATPase-deficient UPF1 mutants (*Figure 4—figure supplement 1B*).

To define the particular RNA context involved in promoting the accumulation of 3′ decay intermediates, a series of single-nucleotide substitutions were introduced into the *GFP^PTC125* reporter at both the precise sites of the 5′ termini and within the region where we predict a stalled ribosome that blocks XRN1 progression would reside on the RNA downstream of the PTC (*Figure 4C*, top). Primer extension analysis of the various reporters in the *UPF1 DE572AA* mutant revealed a number of perturbations in the termini of the 3′ decay fragments (*Figure 4C*). Interestingly, mutation of nucleotides at position +17 or +20, which represent the termini of cDNA ends from the wild-type *GFP^PTC125* reporter, did not alter the accumulation of decay intermediates or the nature of their 5′ termini, demonstrating that the blockage site for 5′ → 3′ decay is neither dependent upon the particular nucleotide present at these positions nor their propensity for base pairing with other RNA residues. By contrast, the relative levels and/or positions of 5′ termini were altered when various substitutions were introduced downstream of the +17 and +20 sites. Specifically, mutation of individual residues between 25 and 34 nt downstream of the PTC altered the relative intensities or caused loss of one of the two cDNA products that are reproducibly generated from the parental reporter mRNA, without noteworthy accumulation of products with 5′ ends at other positions (*Figure 4C*). Critically, the stretch of nucleotides 25–34 nt downstream of the PTC correspond to those expected to be within the mRNA channel (*Ingolia et al., 2009*; *Pelechano et al., 2015*) and near the A, P, and E sites of a ribosome whose footprint would protect up to position +17 (*Figure 4C*, top). These data provide evidence that sequence context has a substantial impact on the position of the ribosome stall site; however, additional experiments will be needed to establish the precise nature of the relationship between nucleotide sequence and ribosome stalling.

Based on our mutational analysis, we anticipated that if a ribosome was indeed stalled downstream of the PTC on our reporter mRNA at a location that would block 5′ → 3′ decay and protect to the +17 position (as illustrated in *Figure 4C*), that decreasing the distance between the nonsense codon and the stall site sufficiently would result in contact between the stalled ribosome and a ribosome terminating at the PTC, and potentially influence accumulation of the 3′ RNA decay fragments. As we had shown above, deletion of three or six nucleotides just downstream of the PTC did not alter the 5′ termini of the 3′ decay fragments as monitored by primer extension (*Figure 4A*). By contrast, deletion of nine nucleotides resulted in a modest reduction in the abundance of the decay

fragment terminating at the +17 position, and deletion of 12 nucleotides caused complete loss in accumulation of this decay intermediate, while maintaining levels of the +20 fragment (*Figure 4D*). These data are consistent with the ability of a terminating ribosome with a footprint that extends 10 nt downstream of the last nucleotide of the premature stop codon (*Ingolia et al., 2009*; *Pelechano et al., 2015*) to collide with and displace a ribosome stalled downstream of the +17 position. Moreover, as would be expected for this configuration, deletion of greater than 12 nucleotides downstream of the PTC led to loss also of the +20 decay fragment (*Figure 4D*). Notably, disappearance of the +17 and +20 fragments did not correspond with accumulation of alternative decay fragments with 5' ends at positions downstream of these sites, suggesting that in this context, the stalled ribosome is displaced upon collision with a terminating ribosome, and does not re-establish a stable interaction with the downstream RNA (*Figure 4D*). Analysis of our additional PTC-containing *GFP* reporters that show variability in sequence context and positioning of stall sites (*Figure 1—figure supplement 1C and D*) will be required to determine the generality of this observation and whether displaced 3' UTR ribosomes can stall at secondary sites in different contexts.

## UPF1 active site mutations alter termini of 3' RNA decay fragments

Using biochemically characterized mutants of yeast UPF1, we previously documented that 3' RNA decay fragments from PTC-containing mRNA accumulate in cells expressing UPF1 which is capable of binding ATP but deficient in catalyzing its hydrolysis (i.e. DE572AA), and not mutants unable to bind ATP (i.e. K436E) (*Serdar et al., 2016*). To extend this observation, additional UPF1 mutants described in the literature as either deficient in in vitro ATP binding (R639A) or ATP hydrolysis (Q601A) activity (*Cheng et al., 2007*) were assessed for fragment formation. Consistent with our earlier findings, 3' RNA decay fragments from $GFP^{PTC125}$ mRNA were evident only in the ATPase-deficient UPF1 Q601A mutant and not the R639A mutant which has been reported to lack detectable ATP binding activity (*Figure 5A*). These findings emphasize allele-specific functional differences between UPF1 active site mutants that are likely due to structural changes predicted to occur during the cycle of ATP binding and hydrolysis (*Cheng et al., 2007*).

To probe the allele-specific function of UPF1 mutants further, we introduced amino acid substitutions into residues within the binding pocket of UPF1 that by structural analysis are predicted to contact ATP (*Chakrabarti et al., 2011*; *Figure 5B*), including several positions that had already been subject to mutational analysis (i.e. D572, E573, and Q601). Substitution of alanine for tyrosine 638, valine 438 or glutamine 413, each of which interact with either the adenine and/or ribose moiety of bound ATP, failed to result in the accumulation of a 3' RNA decay fragment from $GFP^{PTC125}$ mRNA (*Figure 5C*, lanes 5, 11, and 13). Notably, mutation of these positions also did not impair NMD as measured by the ability of cells to reduced steady-state levels of the endogenous substrate, *CYH2* pre-mRNA (*Figure 5—figure supplement 1B*), suggesting that, at least individually, these residues do not play essential roles in UPF1 function (including ATP binding). By contrast, substitution of residues that interact with the gamma phosphate of ATP or coordinate the active site $Mg^{2+}$ ion and which are expected to contribute in catalysis and/or product release all gave rise to 3' decay fragments on northern blots similar in size to that observed for the UPF1 DE572AA mutation (i.e. T437A, E769A, R801A, Q766K, and E769K; *Figure 5B* and *Figure 5C*, lanes 4, 6, 7, 14, and 15). Interestingly, while the majority of these mutations completely inactivated NMD, substitutions at Q766 and E769 lead to only partial impairment of NMD activity (*Figure 5—figure supplement 1B*), likely due to incomplete inhibition of ATP hydrolysis in these mutants. We noted that the intensities of 3' RNA decay fragments differ in these mutants and in comparison to the DE572AA allele; however, these differences did not appear to correlate with NMD activity as measured by *CYH2* pre-mRNA levels (*Figure 5—figure supplement 1B*). Finally, we introduced additional mutations into residues 572/573 and 601 formerly implicated in promoting ATP hydrolysis. Strikingly, in contrast to the alanine substitutions in the DE572AA mutation, introduction of either bulky lysines (DE572KK) or polar asparagine/glutamine (DE572NQ) failed to lead to 3' RNA fragment accumulation (*Figure 5C*, lanes 8 and 9), despite these substitutions resulting in complete abrogation of NMD activity (*Figure 5—figure supplement 1B*). Contrary to this, introduction of lysine at position 601 leads to both a complete loss of NMD activity and a strong accumulation of the 3' RNA fragment (*Figure 5B*, lane 10 and *Figure 5—figure supplement 1B*), similar to what we observe for the mutant in which alanine was introduced at this position (i.e. Q601A).

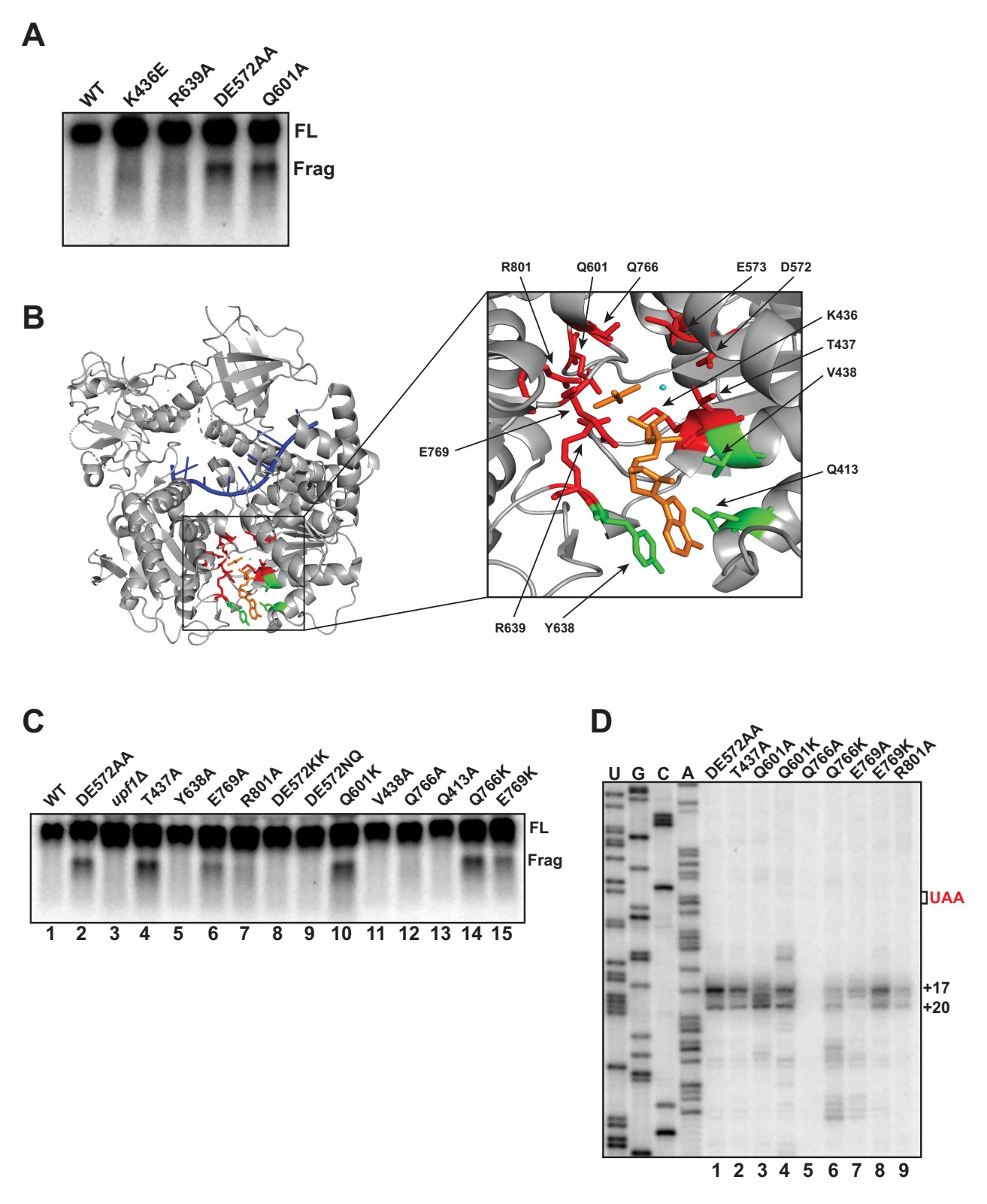

**Figure 5.** Mutational analysis of UPF1 active site correlates ATP hydrolysis with ribosome stalling. Northern blot (**A, C**) and primer extension analysis (**D**) of *GFP^PTC125* mRNA from cells expressing either wild-type UPF1 or the indicated mutant allele with substitutions in the ATP binding pocket. Full-length mRNA (FL) and 3' RNA decay fragments (Frag) are indicated. (**B**) Crystal structure of UPF1 in complex with RNA and ADP:AlF$_4^-$ (PDB accession code 2XZL; *Chakrabarti et al., 2011*). RNA, ADP:AlF$_4^-$ and Mg$^{2+}$ are indicated in blue, orange, and cyan, respectively. Amino acid residues labeled green did

*Figure 5 continued on next page*

*Figure 5 continued*

not give rise to detectable 3′ RNA decay fragments from *GFP^PTC125^* mRNA when mutated; residues in red accumulated 3′ RNA fragments when mutated.

The online version of this article includes the following figure supplement(s) for figure 5:

**Figure supplement 1.** Impact of UPF1 active site mutations on NMD activity.

Differences in the relative intensities of the 3′ RNA decay fragments that accumulate in our various UPF1 active site mutants (*Figure 5C*) and the expectation that these residues contribute to catalysis through distinct mechanisms prompted us to perform primer extension analysis on RNA isolated from cells expressing *GFP^PTC125^* mRNA and several of the *UPF1* variants. As observed in *Figure 5D*, significant differences in the 5′ termini of the decay intermediates were observed for a number of *UPF1* mutant alleles. Most strikingly was the loss or redistribution of termini from the +17 and +20 positions (characteristic of the UPF1 DE572AA mutant) in the Q601A, Q766K, and E769A variants (lanes 3, 6, and 7). Remarkably, these three UPF1 variants also exhibit intermediate activity in targeting a substrate to NMD (*Figure 5—figure supplement 1A and B* and *Cheng et al., 2007*), revealing an important relationship between the efficiency of NMD and the activity of UPF1 on what we perceive as the prematurely terminating ribosome (see Discussion). Given that all three of these residues are positioned to interact either directly or through water bridges with the gamma phosphate of ATP and are expected to contribute to either the rate of ATP hydrolysis or in the release of the phosphate product after cleavage, we introduced additional mutations at these positions. In contrast to the alanine substitution at Q601, lysine at this position resulted in accumulation of *GFP^PTC125^* mRNA fragments with similar termini as the DE572AA mutation but with additional products with 5′ ends both upstream and downstream of these residues (*Figure 5D*; compare DE572AA versus Q601A and Q601K). Interestingly, while replacement of glutamine at position 766 within UPF1 with lysine resulted in distinct cDNA products centered around +28 and +37 nucleotides downstream of the PTC, introduction of alanine at this site caused loss of detectable 3′ RNA decay fragments (*Figure 5D*, lanes 5 and 6). Likewise, although the UPF1 E769A mutant gave rise to 3′ decay products that differed in their 5′ termini compared to the DE mutant, introduction of lysine at this position resulted in intermediates terminating at the characteristic +17 and +20 positions (compare lanes 7 and 8). Taken together, mutational analysis of the UPF1 active site revealed changes relative to DE572AA that altered either 3′ decay fragment accumulation and/or the precise positioning of the 5′ termini of these intermediates. Based on our earlier analyses with biochemically characterized UPF1 mutants (*Figure 5A* and *Serdar et al., 2016*) and structural analysis of the UPF1 active site, these data implicate active site residues involved in catalysis and/or release of the gamma phosphate during ATP hydrolysis in the overall activity of UPF1 in NMD and, critically, the fate of a prematurely terminating ribosome as monitored by the termini of the 3′ decay intermediates that accrue in these mutants.

## Discussion

We have leveraged the 3′ RNA decay intermediates that accumulate from PTC-containing mRNA in the presence of ATPase-deficient UPF1 to enhance our understanding of the functional relationship between the translation and NMD machinery. Our previous characterization of these fragments showed that they were coincident with the transcript 3′ of the PTC, co-sediment with 80S monosomes, and accrue due to a block in digestion by the 5′ → 3′ exoribonuclease XRN1, leading us to conclude that ATP hydrolysis by UPF1 was required for efficient translation termination at a PTC (*Serdar et al., 2016*). High-resolution analysis of these decay intermediates presented here confirm that the fragments are ribosome associated, but reveal, unexpectedly, that the 5′ end of the RNAs map downstream of the PTC, signifying that the impediment to decay is beyond the termination codon and that ribosome(s) occupy the 3′ UTR of these transcripts in the UPF1 mutant. These findings alongside the comparative analysis of RNA decay intermediates that amass upon a genuine block in translation termination (i.e. depletion of eRF1) argue against the simple notion that the efficiency of termination by ribosomes at the PTC is impaired in UPF1 ATPase mutants. Indeed, evidence supporting inefficient translation termination including ribosomal pausing at the PTC or stop

codon read through is lacking (*Figure 1A* and *Figure 3B*), and, in contrast, we observe accumulation of truncated polypeptides in the UPF1 ATPase mutants that by mass spectrophotometric analysis possess a carboxy-terminus expected for accurate peptide hydrolysis at the PTC (*Figure 3B* and data not shown). Our data thereby establish that UPF1 impinges on the translation machinery in a novel manner such that when hydrolysis of bound ATP is blocked, ribosome recycling is inhibited and post-termination ribosomes migrate into the transcript 3' UTR and contribute to hindering XRN1 and preventing full degradation of the transcript downstream of the PTC.

In principle, a number of RNA elements could interfere with XRN1 progression leading to incomplete degradation of the nonsense-containing mRNAs in UPF1 ATPase mutants. RNA structure including thermostable stem-loops (*Garcia and Parker, 2015*), stretches of poly-guanine nucleotides (*Decker and Parker, 1993*), and three-dimensional XRN1-resistant RNA folds (recently identified in the flaviviral RNA genome) (*Chapman et al., 2014*) all obstruct XRN1 and protect downstream RNA from being digested. Inspection and computational modeling of the first 50 nucleotides of the 5' end of the decay fragments that accumulate in our various reporter mRNAs failed, however, to identify common nucleotides/motifs or predict RNA secondary structure that might contribute to impeding XRN1. Moreover, altering the terminal nucleotide of the fragments that accrue from *GFP^PTC125* mRNA did not influence the 5' end of the decay intermediates, as would be expected if they were engaged in base pairing necessary for preventing XRN1 progression (*Figure 4C*). While these data do not strictly rule out the possibility that a thermodynamically stable RNA structural element contributes in blocking XRN1 in our PTC-containing mRNA, we think it unlikely as it would require that such elements exist downstream and proximal of each nonsense mutation we arbitrarily introduced into our various reporter genes in order to lead to the 3' decay intermediates we consistently observe (*Serdar et al., 2016*). Alternatively, an mRNP tightly bound to the mRNA downstream of the PTC could, theoretically, cause inhibition of XRN1 activity. One plausible component of such an mRNP would be the ATPase-deficient UPF1 protein itself, which displays increased association with both NMD target and non-target RNAs compared to wild-type UPF1 (*Kurosaki et al., 2014*; *Lee et al., 2015*). Although we anticipate that UPF1 associates within the 3' UTR of our PTC-containing transcripts (*Hurt et al., 2013*; *Kurosaki and Maquat, 2013*; *Zünd et al., 2013*), the expected footprint for RNA-bound UPF1 is ~11 nucleotides (*Chakrabarti et al., 2011*), making it difficult to explain how changes within *GFP^PTC125* 12–18 nucleotides downstream of the position of the 5' end of the decay intermediate could impact UPF1 binding so as to alter fragment accumulation (*Figure 4C*). Moreover, any model in which RNA-bound UPF1, alone or in complex with other proteins, serves as the block to XRN1 activity would have to account for how a ribosome can traverse this block to establish the demonstrated association with the 3' RNA fragment (*Figure 2B*).

In contrast, evidence supports a view in which the impediment to complete digestion of the PTC-containing mRNA by XRN1 is the 3' UTR-bound ribosome itself. The ability of a ribosome to interfere with XRN1 progression is well documented from co-translational decay studies employing reporters harboring rare codons designed to slow translation elongation (*Hu et al., 2009*) and in global characterization of XRN1-dependent mRNA decay intermediates (*Pelechano et al., 2015*). We demonstrate that the 3' RNA decay fragments in UPF1 ATPase mutants physically associate with ribosomes and that mutation of residues expected to be accommodated within the mRNA binding channel of a ribosome protecting the 5' terminus of the decay fragment strongly influence the accumulation of the intermediates (*Figure 4C*). Attempts to gather direct evidence for ribosome occupancy at the 5' end of the decay fragment employing ribosome profiling have, however, been unsuccessful. Although Ribo-Seq libraries from UPF1 ATPase mutants assigned ribosomes to the coding region of *GFP^PTC125* mRNA at high density and with periodicity, ribosome protected fragments (RPFs) mapping to the 3' UTR of the transcript were absent from our datasets. As discussed further below, we speculate that the 3' UTR-bound ribosome(s) are inherently instable and/or insensitive to cycloheximide, which would contribute to omission of RPFs mapping to the 3' UTR from our preparations; although it is also possible that the 3' UTR-associated ribosomes assume a confirmation that protects RNA greater or less than ~28 nucleotides, and were unintentionally precluded from our libraries.

Together, our observations indicate that 80S ribosomes were not released from mRNA after peptide hydrolysis at the PTC and suggest that post-termination ribosome recycling is inhibited in UPF1 mutants defective in ATPase activity. Recent ribosome profiling in yeast suggests that movement of post-termination ribosomes into mRNA 3' UTRs is rare but markedly enhanced in cells depleted of

ribosome recycling factor RLI1 (*Young et al., 2015*). RLI1 (ABCE1 in mammals) is an Fe-S cluster protein that has been implicated in at least two steps of translation termination, including promoting peptide hydrolysis by eRF1 in an ATPase-independent manner and driving ribosomal subunit separation in a process that requires ATP hydrolysis (*Dever and Green, 2012*). Consistent with this, in RLI-deficient yeast cells, where both steps would be prevented, ribosomes are found to both queue at stop codons and transit downstream into the mRNA 3′ UTR where they appear competent to reinitiate translation without an apparent reading frame or start codon preference (*Young et al., 2015*). Although we provide evidence for 3′ UTR-bound ribosomes downstream of the PTC in the *UPF1 DE572AA* mutant, our data is not consistent with ATPase-deficient UPF1 simply preventing RLI1 from interacting with the terminating ribosome (akin to depleting it from the cell). Indeed, we showed previously that 3′ decay fragments from *GFP^PTC125* mRNA in cells with reduced cellular RLI1 levels display greater size heterogeneity on northern blots than in the UPF1 mutant (*Serdar et al., 2016*) and evidence for 3′ UTR translation products or a queuing of ribosomes at the stop codon is lacking in UPF1 ATPase mutants (*Figure 3B* and *Figure 1A*). Then how could a defect in ATP hydrolysis by UPF1 lead to 3′ UTR ribosomes? One model that reconciles our data posits that mutant UPF1 inhibits RLI1 function not during recruitment to the terminating ribosome or in promoting peptide hydrolysis, but subsequently in its ability to catalyze ATP-dependent ribosome subunit dissociation, leading to ribosome migration into the mRNA 3′ UTR (*Figure 6*). In such a scenario, ribosomes blocked during the second step would be expected to retain eRF1 and catalytically-stalled RLI1 in the A site and thus be incompetent for canonical translation elongation into the mRNA 3′ UTR and any reinitiation of protein synthesis. Consistent with this model and a block in RLI1 function after its association with ribosomes terminating at the PTC, overexpression of RLI1 in UPF1 ATPase mutants does not abrogate accumulation of 3′ decay fragments, and deletion of DOM34, which binds within

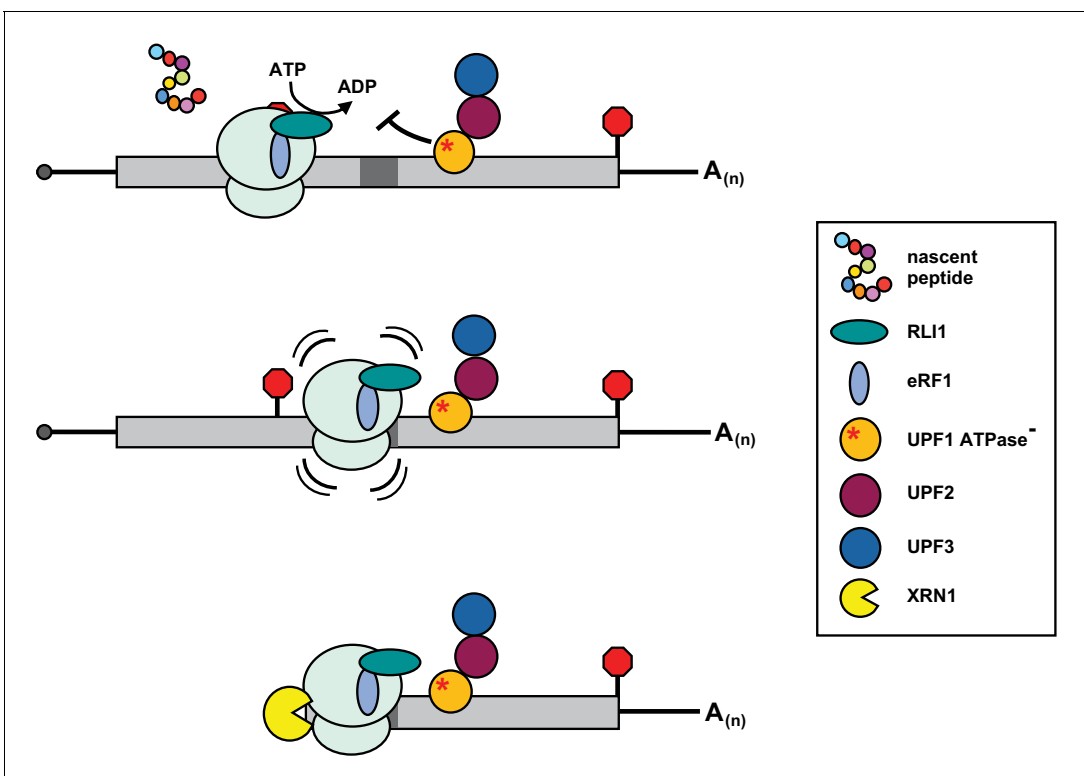

**Figure 6.** Inhibition of post-termination ribosome release during premature translation termination in UPF1-ATPase mutants. RLI1 function in translation termination involves both stimulation of peptidyl-tRNA hydrolysis by eRF1 and ATP hydrolysis-dependent ribosome subunit splitting. In UPF1 ATPase mutants, ribosomes efficiently recognize a PTC and release the C-terminally truncated polypeptide; however, ribosome subunits fail to be dissociated from the RNA. Post-termination ribosomes with an occupied A site migrate downstream of the PTC into the mRNA 3′ UTR where they stall in a manner dependent upon RNA sequence (dark gray box) and the nature of the inactivating mutation in the ATP binding pocket of UPF1, and block 5′ → 3′ degradation of the mRNA by XRN1.

the ribosome A site to promote rescue of stalled ribosomes (*Guydosh and Green, 2014*), does not lead to an increase in the accumulation of 3' decay intermediates (data not shown). Moreover, because RLI1-bound 80S ribosomes would be incapable of re-engaging in translation, they would be insensitive to stabilization by cycloheximide, and likely lost during the sucrose gradient centrifugation step of our ribosome profiling protocol.

Our analysis of several PTC-containing reporters reveals that 3' RNA decay intermediates differ in their heterogeneity and in the position of the 5' ends relative to the PTC. We found that site selection for ribosome stalling was dependent upon the sequence downstream of the PTC (*Figure 4B*) but we were unable to ascertain common RNA features that contribute to establishing the RNA 5' end. Specifically, we did not observe the consistent presence of an AUG codon in the predicted P site of a ribosome stalled at the 5' end of the decay intermediates, as has been observed for ribosomes detected upstream of a PTC following premature termination in in vitro translation extracts (*Amrani et al., 2004*). Notably, for $GFP^{PTC125}$ mRNA, which does encode an AUG triplet proximal to the predicted P site, mutation of nucleotides within the AUG did alter the distribution and/or intensity of 5' ends of the decay intermediates but as did substitution of several non-AUG residues flanking this triplet (*Figure 4C*). In addition, our finding that the identity of the 5' ends of decay fragments is independent of the nature of the penultimate codon upstream of the PTC (*Figure 4— figure supplement 1B*) rules out a role for base complementarily between the mRNA and a P site tRNA that might be retained post termination. One intriguing observation is that the 5' termini of the decay fragments in UPF1 ATPase mutants are sensitive to allele-specific amino acid substitutions predicted to influence ATP hydrolysis or retention of either the substrate or product within the enzyme active site (*Figure 5B and D*). Although these data closely link the catalytic function of UPF1 with ribosome dynamics post termination, the precise determinants dictating ribosome stalling remain unclear, and likely are determined by a combination of the kinetics of ATP hydrolysis with sequence elements, mRNP composition, and/or RNA structural landscape that is unique to each mRNA 3' UTR.

We provide evidence supporting a model in which ATPase-deficient UPF1 inhibits prematurely terminating ribosome release resulting in migration of 80S ribosomes downstream of the PTC where they stall and block XRN1-mediated degradation of the mRNA. In the context of NMD in wild-type cells, our data are supportive of a function for UPF1 in which conformational changes to the protein induced by ATP hydrolysis promotes RLI1 ATP-dependent ribosome subunit dissociation at a PTC. It is tempting to speculate that modulation of this latter step in the translation termination process could serve as the molecular indicator that is sensed and transmitted to the cellular mRNA decay machinery leading to the accelerated degradation characteristic of NMD substrates. This proposed coupling of ribosome dynamics with changes in transcript stability for NMD substrates aligns with an emerging theme exemplified by modulation of mRNA decay by codon identity and translation elongation rate (*Radhakrishnan and Green, 2016*), as well as in the detection and rescue of stalled ribosomes (*Ikeuchi et al., 2018*). Future research will be needed to define the precise nature of the perturbation in RLI1 function by the NMD machinery during premature translation termination. Such studies will provide an exciting opportunity to biochemically differentiate translation termination events specific to nonsense codons and for the development of therapeutic agents capable of modulating premature termination with high specificity.

# Materials and methods

**Key resources table**

| Reagent type (species) or resource | Designation | Source or reference | Identifiers | Additional information |
|---|---|---|---|---|
| Strain, strain background (*Saccharomyces cerevisiae*) | Wild type (WT) | Saccharomyces Genome Deletion Project | | MATa, *ura3, leu2, his3, met15* |

*Continued on next page*

*Continued*

| Reagent type (species) or resource | Designation | Source or reference | Identifiers | Additional information |
|---|---|---|---|---|
| Genetic Reagent (*S. cerevisiae*) | *upf1Δ* | Saccharomyces Genome Deletion Project | | MATa, *ura3, leu2, his3, met15*, upf1::KanMX |
| Genetic Reagent (*S. cerevisiae*) | Rpl16-ZZ | This paper | | MATa, *ura3, leu2, his3, met15*, upf1::KanMX, RPL16A-ZZ-HIS3 |
| Genetic Reagent (*S. cerevisiae*) | Rps13-HA | This paper | | MATa, *ura3, leu2, his3, met15*, upf1::KanMX, RPS13-HA-HIS3 |
| Genetic Reagent (*S. cerevisiae*) | Sup45 depletion strain | This paper | | MATa, *ura3, leu2, his3, met15*, upf1::KanMX, $HIS3$-$P_{GAL}$-3HA-SUP45 |
| Antibody | Anti-HA (Mouse monoclonal) | Covance | MMS-101P; RRID:AB_2314672 | WB: (1:5,000) IP: (4 μg) |
| Antibody | Anti-TAP (Rabbit polyclonal) | Thermo Fisher | CAB1001; RRID:AB_10709700 | IP: (4 μg) |
| Antibody | Anti-Pab1 (Mouse monoclonal) | Encore Biotechnology | MCA-1G1; RRID:AB_2572370 | WB: (1:10,000) |
| Antibody | Anti-mouse IgG-HRP (goat polyclonal) | Santa Cruz Biotechnology | Sc-2005; RRID:AB_631736 | WB: (1:5,000) |
| Commercial Assay or Kit | Sequenase 2.0 DNA Sequencing Kit | Thermo Fisher | 70771KT | |
| Recombinant DNA Reagent | $GFP^{PTC67}$ | This paper | pKB673 | CEN; URA3 |
| Recombinant DNA Reagent | $GFP^{PTC125}$ | This paper | pKB674 | CEN; URA3 |
| Recombinant DNA Reagent | $GFP^{PTC135}$ | This paper | pKB510 | CEN; URA3 |
| Recombinant DNA Reagent | *UPF1-WT* | PMID:28008922 | pKB556 | CEN; URA3 |
| Recombinant DNA Reagent | *UPF1-DE572AA* | PMID:28008922 | pKB576 | CEN; LEU2 |
| Sequence-based reagent | RT primer (*$GFP^{PTC125}$* primer extension analysis) | This paper | oKB132 | GGGCAGATTGTGTGGACAGGTAATGGTTGTCTGG |

## Yeast culture

A complete list of yeast strains used in this study is provided in *Supplementary file 1*. Yeast cultures were grown at 30°C with shaking at 250 RPM in synthetic medium supplemented with appropriate amino acids and either 2% glucose (SD) or 2% galactose and 1% sucrose (SGS).

## Plasmid construction

Plasmids were generated using standard molecular cloning techniques and by site-directed PCR mutagenesis. A complete list of plasmids and oligonucleotides used in this work is provided in *Supplementary file 2* and *Supplementary file 3*, respectively.

## RNA isolation and northern blot analysis

Yeast cultures (50 mL) were grown to mid-log phase and flash frozen on dry ice. Total cellular RNA was isolated from frozen cell pellets using glass bead disruption and phenol-chloroform extraction. 30 µg of RNA was analyzed on 1.4% agarose gels containing 5.92% formaldehyde. RNA was transferred to Hybond–N nylon membrane (GE Healthcare) and UV crosslinked. Membranes were washed in $0.1\times$ SSC/0.1% SDS buffer before probing overnight with radiolabeled oligonucleotides.

## Ribosome affinity purification

Cells expressing epitope-tagged RPL16A or RPS13 (*Supplementary file 1* Table S1) were grown in 250 mL cultures to mid-log phase and flash frozen on dry ice. Cells were lysed at 4°C in polysome lysis buffer (10 mM Tris pH 7.4, 100 mM NaCl, 30 mM $MgCl_2$, 1 mM DTT, 100 µg/mL cycloheximide) using glass bead disruption. Lysates were cleared by centrifugation at 2000 rpm for 2 min. About 10–20 $OD_{600}$ units of cell lysate were incubated with either α-HA (Covance; MMS-101P) or α-TAP (Thermo Fisher; CAB1001) antibodies for 1 hr at 4°C with gentle rocking. Antibody-bound lysates were then incubated with Protein G Dynabeads (Thermo Fisher; 1004D) for an additional 60 min at 4°C. Beads were washed three times in IXA-100 buffer (50 mM Tris-HCl pH 7.5, 100 mM KCl, 12 mM Mg(OAc), 1 mM DTT, 100 µg/mL cycloheximide). RNA was eluted at 95°C in elution buffer (50 mM Tris-HCl pH 7.5, 0.5% SDS, 50 mM EDTA), phenol-chloroform extracted, and resuspended in LET buffer for northern blot or primer extension analysis.

## Protein isolation and western blot analysis

Yeast cultures (50 mL) were grown to mid-log phase and flash frozen on dry ice. Cell pellets were heated in 5 M urea for 2 min at 95°C and lysed by mechanical disruption with glass beads by vortexing for 5 min. Solution A (125 mM Tris-HCl, pH 6.8, 2% SDS) was added to lysates and samples vortexed for 1 min followed by heating to 95°C for 2 min. Glass beads and cellular debris were cleared from lysates by centrifugation at 13,200 RPM for 4 min. Equivalent units ($OD_{260}$) of cell lysate in $1\times$ SDS sample buffer (125 mM Tris-HCl, pH 6.8, 2% SDS, 100 mM DTT, 10% glycerol, 0.05% bromophenol blue) were separated on 7.5% Bis-Tris polyacrylamide gels by electrophoresis in $1\times$ SDS running buffer (25 mM Tris base, 192 mM glycine, 0.1% SDS). Proteins were transferred to PVDF transfer membrane (Thermo Fisher) in $1\times$ transfer buffer (25 mM Tris base, 192 mM glycine, 20% methanol) by electroblotting at 4°C for 2 hr at 250 mA. Membranes were blocked (5% milk powder in $1\times$ TBS/0.1% Tween-20) overnight at 4°C and proteins detected by incubating with primary antibodies [mouse monoclonal α-HA 1:5000 (Covance; MMS-101P) or mouse monoclonal α-PAB1 1:10,000 (Encore Biotechnology; MCA-1G1)] and secondary antibodies [goat α-mouse IgG HRP 1:5000 (Santa Cruz Biotechnology; sc-2005)] in blocking buffer for 1 hr at room temperature. Between incubations, membranes were washed with $1\times$ TBS/0.1% Tween-20 three times each for 15 min. Signals were detected using chemiluminescence with Blue Ultra Autorad film (GeneMate).

## Conditional depletion of *SUP45*

Yeast strains for depletion studies were constructed by placing chromosomally-encoded *SUP45* under control of the *GAL1* promoter using standard recombinant methods (*Longtine et al., 1998*). Briefly, a His3M $\times$ 6-$P_{GAL1}$-3HA insertion cassette was PCR-amplified using Phusion High Fidelity DNA polymerase (NEB; M0530S) and gene-specific PCR primers (*Supplementary file 3* Table S3). PCR products were run on 1% agarose gels, purified using Zymoclean Gel DNA Recovery Kits (Zymo Research; D4001), and transformed into wild type (yKB154), or *upf1Δ* (yKB146) yeast strains. For depletion experiments, cells were inoculated at an initial density of $OD_{600} = 0.02$ in SGS media and grown at 30°C for 12 hr, cells were then pelleted by centrifugation at 4000 RPM for 4 min, washed once in synthetic media without sugar, and resuspended in glucose-containing media (SD) to an $OD_{600} = 0.1$. Cultures were grown at 30°C for 10 hr and aliquots collected every 2 hr. The cell pelleted after centrifugation was flash frozen on dry ice for downstream protein analysis and reporter mRNA expression using western and northern blotting, respectively.

## Primer extension analysis

Phenol-chloroform extracted total RNA was reverse transcribed using SuperscriptIII Reverse Transcriptase (Thermo Fisher; 18080051) and $^{32}P$ end-labeled primers specific to each reporter mRNA

(*Supplementary file 3* Table S3). Dideoxy sequencing reactions were performed using the Sequenase 2.0 DNA Sequencing Kit (Thermo Fisher; 70771KT). Sequenase reactions were performed using the plasmid encoding the reporter mRNA (pKB674 for $GFP^{PTC125}$) as a template and primed with the same $^{32}$P end-labeled primer as was used for reverse transcription (oKB132 for $GFP^{PTC125}$ mRNA). Primer extension products and sequencing reactions were run on 6% PAGE-UREA denaturing gels that were fixed in 10% acetic acid/10% methanol solution for 15 min and dried under vacuum pressure before exposure to phosphor screens.

## Acknowledgements

The authors would like to thank Tim Nilsen for helpful comments on this work.

## Additional information

### Funding

| Funder | Grant reference number | Author |
|---|---|---|
| National Institute of General Medical Sciences | GM095621 | Kristian Baker |
| National Science Foundation | 1253788 | Kristian Baker |
| National Institutes of Health | GM08056 | Lucas D Serdar |

The funders had no role in study design, data collection and interpretation, or the decision to submit the work for publication.

### Author contributions

Lucas D Serdar, Formal analysis, Validation, Investigation, Writing - original draft; DaJuan L Whiteside, Formal analysis, Validation, Investigation; Sarah L Nock, Investigation; David McGrath, Formal analysis, Investigation; Kristian E Baker, Conceptualization, Resources, Formal analysis, Supervision, Funding acquisition, Project administration, Writing - review and editing

### Author ORCIDs

Lucas D Serdar (iD) https://orcid.org/0000-0002-4574-1668
Kristian E Baker (iD) https://orcid.org/0000-0003-2262-5434

### Decision letter and Author response

Decision letter https://doi.org/10.7554/eLife.57834.sa1
Author response https://doi.org/10.7554/eLife.57834.sa2

## Additional files

### Supplementary files

- Supplementary file 1. Complete list of all yeast strains used in this study.
- Supplementary file 2. Complete list of all plasmids used in this study.
- Supplementary file 3. Complete list of all oligonucleotides used in this study.
- Transparent reporting form

### Data availability

All data generated or analysed during this study are included in the manuscript and supporting files.

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
