## [Decision Letter]

**Acceptance summary:**

This work is significant in providing evidence that genetically impairing the ATPase activity of UPF1, a factor central to the process of nonsense-mediated mRNA decay (NMD), interferes with the process of ribosome recycling by ABCE1/RLI1 following translation termination at a premature termination codon, allowing the unrecycled 80S ribosomes to migrate into the 3' UTR and stall at particular sequences. As such, it represents a valuable contribution to our understanding of the influence of NMD proteins on the process of translation termination at premature stop codons.

**Decision letter after peer review:**

Thank you for submitting your article "Inhibition of post-termination ribosome recycling at premature termination codons in UPF1 ATPase mutants" for consideration by *eLife*. Your article has been reviewed by three peer reviewers, one of whom is a member of our Board of Reviewing Editors, and the evaluation has been overseen by Alan Hinnebusch as the Reviewing Editor and James Manley as the Senior Editor. The following individual involved in review of your submission has agreed to reveal their identity: Oliver Mühlemann (Reviewer #3).

The reviewers have discussed the reviews with one another and the Reviewing Editor has drafted this decision to help you prepare a revised submission.

We would like to draw your attention to changes in our revision policy that we have made in response to COVID-19 (https://elifesciences.org/articles/57162). Specifically, when editors judge that a submitted work as a whole belongs in *eLife* but that some conclusions require a modest amount of additional new data, as they do with your paper, we are asking that the manuscript be revised to either limit claims to those supported by data in hand, or to explicitly state that the relevant conclusions require additional supporting data. If you choose to pursue this option and re-submit the manuscript without performing the requested additional experiments, and you cannot convince the reviewers that they are not required to justify your main claims, then you will be committing yourself to eventually carrying out the additional experiments and reporting on how they affect the relevant conclusions either in a preprint on bioRxiv or medRxiv, or if appropriate, as a Research Advance in *eLife*, either of which would be linked to the original paper.

Summary:

In this report, the authors study decay intermediates from a reporter mRNA harboring a PTC that accumulate in a UPF1 ATPase mutant, but not in the absence of UPF1 protein, and which were shown previously to originate from inhibition of XRN1 5' to 3' exonucleolytic decay by ribosomes on the mRNA. Here they show that the 5' ends of these intermediates map downstream of the PTC rather than coinciding with the PTC itself as had been proposed previously. Thus, rather than providing evidence of a ribosome stalled at the PTC owing to inefficient termination; they instead appear to represent an unrecycled 80S ribosome that has migrated from the PTC and become stalled at specific positions downstream in the UPF1 ATPase mutant.

Coimmunoprecipitation analysis of tagged Rps13 shows that these decay fragments are ribosome bound. Other evidence suggests that the protecting ribosomes are not engaged in either translational read-through of the PTC, nor from reinitiation of translation 3' of the PTC. Evidence further indicates that the distance from the PTC is not a critical factor in determining the site of the block to 5'-3' decay presumed to arise from the stalled 80S ribosome; and instead, the nucleotide sequence following the PTC determines the site of presumed stalling and XRN1 inhibition. Other experiments show that the sequence at the site of XRN1 inhibition is not essential for inhibition, but that the accumulation of the decay intermediates is reduced by mutations that would affect mRNA contacts with the predicted stalled ribosome, and can be completely prevented if the distance between the PTC and the stalled site is reduced to the point where the presumed stalled ribosome would collide with the termination complex at the PTC. The latter finding provides some of the best evidence that XRN1 is inhibited by a stalled ribosome rather than by an RNA-binding protein, as the latter would have a much smaller footprint on the mRNA. They further show that the nature of the UPF1 mutation strongly influences the position of the putative stalled ribosome downstream of the PTC, as only mutations that impair ATP hydrolysis and not ATP binding yield the XRN1-resistant decay products, and various active site mutants affecting hydrolysis in different ways produce somewhat different protection patterns. They propose that when hydrolysis of bound ATP is blocked by particular UPF1 mutations, ribosome recycling by ABCE1/RLI1 following termination at the PTC is inhibited and post-termination 80S ribosomes subsequently migrate (without translating) downstream, stall at particular sequences, and thereby block 5'-3- degradation by XRN1. They further propose that in wild-type cells, UPF1 normally promotes RLI1 function in ribosome recycling in a manner dependent on conformational changes to the protein induced by ATP hydrolysis.

Essential revisions:

1) Perform additional experiments to bolster the conclusion that "upon collision with a terminating ribosome, the stalled ribosome is displaced and does not re-establish a stable interaction with the downstream RNA", for example, by constructing deletions with increasing length as in Figure 4D for two other PTCs.

2) Provide control experiments using primer extension to confirm that the protected fragments are not detected downstream of a normal termination codon and are uniquely found downstream of PTCs.

3) Revise the manuscript to address as many as possible of the other major and comments or raised by the reviewers.

Reviewer #1:

In this report, the authors study decay intermediates from a reporter mRNA harboring a PTC that accumulate in a UPF1 ATPase mutant, but not in the absence of UPF1 protein, and which were shown previously to originate from inhibition of XRN1 5' to 3' exonucleolytic decay by ribosomes on the mRNA. Here they show that the 5' ends of these intermediates map downstream of the PTC rather than coinciding with the PTC itself as had been proposed previously. Thus, rather than providing evidence of a ribosome stalled at the PTC owing to inefficient termination; they instead appear to represent an 80S ribosome that has migrated from the PTC and become stalled at specific positions downstream in the UPF1 ATPase mutant. CoIP analysis of tagged Rps13 shows that these decay fragments are ribosome bound. Other evidence suggests that the protecting ribosomes are not engaged in either translational read-through of the PTC, or from reinitiation of translation 3' of the PTC. Their results further indicate that distance from the PTC is not a critical factor in determining the site of the block to 5' -3' decay presumed to arise from the stalled 80S ribosome; and instead, the nucleotide sequence immediately following the PTC determines the site of presumed stalling and XRN1 inhibition. Other experiments show that the sequence at the site of XRN1 inhibition is not essential for inhibition, but that the accumulation of the decay intermediates is reduced by mutations that would affect mRNA contacts with the predicted stalled ribosome, and can be completely prevented if the distance between the PTC and the stalled site is reduced to the point where the presumed stalled ribosome would collide with the termination complex at the PTC. The latter finding provides some of the best evidence that XRN1 is inhibited by a stalled ribosome rather than by an RNA-binding protein, as the latter would have a much smaller footprint on the mRNAs. They further show that the nature of the UPF1 mutation strongly influences the position of the putative stalled ribosome downstream of the PTC, as only mutations that impair ATP hydrolysis and not ATP binding yield the XRN1-resistant decay products, and various active site mutants affecting hydrolysis in different ways produce somewhat different protection patterns. They propose that when hydrolysis of bound ATP is blocked by particular UPF1 mutations, ribosome recycling by ABCE1/RLI1 following termination at the PTC is inhibited and post-termination 80S ribosomes subsequently migrate (without translating) downstream, stall at particular sequences, and thereby block 5'-3- degradation by XRN1. They further propose that in wild-type cells, UPF1 normally promotes RLI1 function in ribosome recycling in a manner dependent on conformational changes to the protein induced by ATP hydrolysis.

The experiments are carefully designed and well-executed and the paper is well written. The results largely justify the model that the ATPase-defective UPF1 mutants somehow interfere with ribosome recycling by RLI1 following termination at the PTC, allowing the unrecycled 80S ribosomes to migrate into the 3'UTR and stall at particular sequences. While it is reasonable to propose that wild-type UPF1 would stimulate recycling by RLI1, it's unclear why UPF1 mutants defective for ATP binding (vs. hydrolysis), nor the absence of UPF1, do not give rise to scanning unrecycled 80S ribosomes that stall at the same sequences 3' of the PTC. It's also not clear whether the proposed activity of UPF1 in stimulating RLI1 recycling function is important for NMD; although it would be surprising if it were not.

1) It seems feasible to test the model that ribosome recycling by ABCE1/RLI1 is impaired by the UPF1 ATPase mutants by determining whether depleting RLI1 from WT UPF1 cells, using a published degron mutant, gives rise to the same reporter decay intermediates.

2) It would enhance the story if they were able to map the 3' ends of the presumed stalled 80S ribosomes downstream of the PTC in order to confirm protection of 3'UTR mRNA sequences by entities with a ribosome-size footprint. They tried conventional ribosome profiling to no avail, but perhaps crosslinking could be attempted in an effort to stabilize the putative scanning ribosome.

Reviewer #2:

This manuscript from the Baker lab describes detailed characterization of RNA decay intermediates arising from expression of ATPase-deficient mutants of UPF1 in *S. cerevisiae*. Building on their earlier work, they find that expression of UPF1 that can bind but not hydrolyze ATP causes accumulation of 80S ribosome-bound fragments with 5' termini at various sites downstream of NMD-inducing termination codons. The authors use several reporters to map the fragment 5' ends, finding that a set of generally well-defined fragments was generated from each reporter, in a manner dependent on the sequence context downstream of the termination codon. While overall rules governing this process remain unclear, they present compelling data that residues positioned at the E/P/A sites of the ribosome are the major determinants of 5' end position. Further, in contrast to various manipulations that alter translation termination and ribosome recycling, they find no evidence that the 80S ribosomes bound downstream of TCs engage in reinitiation or readthrough. Many questions remain, but this work presents convincing evidence that ATPase-deficient UPF1 can impinge on the process of ribosome recycling downstream of PTCs. For that reason, it is a valuable contribution to the field's understanding of roles of NMD proteins at the terminating ribosome.

Due to the current COVID-19 situation, I am focusing on whether the authors’ claims are substantiated by the existing evidence and am limiting my requests to those which do not require further experimentation.

Essential revisions:

1) Figure 4D: Based on deletion analyses which result in the disappearance of bands corresponding to decay fragments, without corresponding appearance of a band elsewhere, the authors conclude that, "upon collision with a terminating ribosome, the stalled ribosome is displaced and does not re-establish a stable interaction with the downstream RNA." I think the general applicability of this conclusion is unclear without more information about why some sequences do not support robust fragment generation. What if, for example, a reporter was constructed in which the sequence used in Figure 1—figure supplement 1D that generates a +31 fragment (with additional bands at +43 and +51) was inserted downstream of the 15 nt deletion reporter sequence? To be clear, I'm not asking for this experiment to be completed-just offering it as an illustration that alternatives to the stated conclusion are possible.

2) I suggest moving Supplementary Figure 3A to the main text so all assays of protein expression in the experiment in Figure 3B can be evaluated as a whole. Likewise, incorporating the model and RNA abundance measurements from Supplementary Figure 5 to the main Figure 5 would make it easier to interpret the data currently in Figure 5. Also, measures of error and stats should be provided if possible.

3) Figure 4A: I think the statement that "the nucleotide distance from the PTC is not a critical factor in determining how XRN1 progression is impeded" is not fully supported. Based on the overall data presented, position likely plays some role, as the products characterized are in a small range downstream of the TC and it is clear that certain positions are incompatible with cleavage or protection of the resulting fragment. It would be better to state that fragment accumulation is likely determined by a combination of sequence and position.

Reviewer #3:

This manuscript represents a follow-up of the work presented in Serdar et al., 2016, in which the showed that expressing of an ATPase-deficient UPF1 mutant in yeast caused the accumulation of 3´RNA decay fragments of NMD-sensitive RNAs whose size was coincident with a block of XRN1-dependent 5´-to-3´exonucleolytic degradation around the PTC position. Based on their data, they concluded that UPF1-mediated ATP hydrolysis was required for ribosome disassembly at PTCs.

Here they now used primer extension assays to map the 5´end of these RNA fragments originating from a *GFP^PTC125^* reporter gene and detect two bands 20 and 17 nucleotides downstream of the PTC (Figure 1). The 5´ ends of the decay fragments from *GFP* reporters with PTCs in two different positions also mapped downstream of the PTC, but with different distances and band patterns: *GFP^PTC67^* showed bands around +20 and +12, and *GFP^PTC135^* many bands at different positions between +51 and +9 (Figure 1—figure supplement 1). A control I very much would like to see here: Are such fragments also detected downstream of the normal *GFP* stop codon in the *GFP* WT reporter when UPF1 fails to hydrolyze ATP? In other words, is this phenomenon PTC-specific?

Immunoprecipitation of tagged ribosomes co-precipitated mainly full-length reporter RNA but also the 3´decay fragment in cell expressing the UPF1 *DE572AA* mutant, suggesting that these RNA fragments are associated with ribosomes (Figure 2 and Figure 2—figure supplement 1).

How do the ribosomes get to these positions downstream of the stop codon? The authors could show that this seems not to occur due to readthrough, because (i) no readthrough was detected in cells expressing the UPF1 mutant and (ii) introducing a frameshift did not change the pattern of these bands (Figure 3 and Figure 3—figure supplement 1). The mapped 5´ends of these 3´decay fragments do neither occur at a specific distance from the PTC nor depend on the identity of the stop codon but rather depend on the sequence context downstream of the PTC (Figure 4 and Figure 4—figure supplement 1).

While the experiments shown in Figure 4 and Figure 4—figure supplement 1 aim at gaining some mechanistic insight into how these XRN1-resistant decay fragments might arise, they appear to stop halfway. To corroborate the conclusion that the sequence context is important, the single nucleotide substitution approach (Figure 4C) should be extended to the other two PTCs and critical residues and some sort of a "consensus sequence" might be revealed by comparing the results from all three contexts. Similarly, deletions with increasing length as in Figure 4D should also been tested for the other two PTCs to further corroborate the hypothesis that upon collision with the terminating ribosomes positioned at the PTC, these 3´ fragments disappear.

Figure 5 finally further corroborates a finding already reported in the previous paper: UPF1 mutants impaired in ATP binding do not cause the appearance of the 3´ fragments, while mutants impaired in ATP hydrolysis do. Interestingly, among the many UPF1 mutants tested, including those that promote rather than inhibit ATP hydrolysis, there was no correlation between appearance of the 3´ fragment and impairment of NMD. Thus, we do not really gain more insight with these mutants regarding the mechanism by which the 3´decay fragments accrue.

In summary, the authors convincingly document the accumulation of ribosome-associated 3´ RNA decay fragments in cells expressing ATPase-deficient UPF1 whose 5´ends map downstream of the stop codon, but the fail to provide much mechanistic insight for this unexpected finding. The presented model is plausible but remains quite speculative without further data explaining the proposed connection between UPF1 ATP hydrolysis and RLI1's function in splitting the ribosomal subunits.

---

## [Author Response]

Essential revisions:1) Perform additional experiments to bolster the conclusion that "upon collision with a terminating ribosome, the stalled ribosome is displaced and does not re-establish a stable interaction with the downstream RNA", for example, by constructing deletions with increasing length as in Figure 4D for two other PTCs.

We appreciate this suggestion and admit that we did not perform deletion analysis on the 3’ UTRs of additional PTC-containing mRNA reporters to address whether decay intermediates are also lost when the position of the 5’ end is brought within the footprint of a ribosome positioned at the PTC. We felt that the step-wise loss of the two primer extension (PE) products for *GFP^PTC125^* mRNA shown in Figure 4D was particularly convincing and clearly showed that the impediment residing at the 5’ end of the decay intermediate that blocks XRN1 progression is eliminated when this position is within 12 nucleotides of the PTC (with no evidence for a new site for XRN1 blockage arising downstream).

We proposed the most parsimonious explanation of our data and believe that our conclusion is well justified, and point out that reviewer 1 appears to agree – *‘…accumulation of the decay intermediates is reduced by mutations that would affect mRNA contacts with the predicted stalled ribosome, and can be completely prevented if the distance between the PTC and the stalled site is reduced to the point where the presumed stalled ribosome would collide with the termination complex at the PTC. The latter finding provides some of the best evidence that XRN1 is inhibited by a stalled ribosome rather than by an RNA-binding protein, as the latter would have a much smaller footprint on the mRNAs’*. Notwithstanding, we have revised the text associated with this result to reflect that additional experiments are needed to confirm the generality of this conclusion.

2) Provide control experiments using primer extension to confirm that the protected fragments are not detected downstream of a normal termination codon and are uniquely found downstream of PTCs.

We agree that this experiment represents a critical control to show ATP-deficient UPF1 impinges exclusively on translation termination events that are ‘premature’. We direct the reviewers to our previous work characterizing this UPF1 mutant (Serdar et al., 2016) where we performed this precise experiment. We evaluated whether 3’ RNA fragments accumulated downstream of a normal translation termination site in UPF1 ATPase mutants using a reporter lacking a PTC, and did *not* detect such fragments (see Figure 1C in Serdar et al.). Based on that result, we concluded that accumulation of the 3’ RNA fragments (that we later showed were ‘decay intermediates’) are dependent on premature translation termination. Admittedly, in that experiment, RNA from the non-PTC containing reporter were analyzed by Northern blot and not primer extension, but notably, we have never detected cDNA products using primer extension that were not previously revealed by Northern blot. Notwithstanding, those data show that 3’ RNA decay fragments do not accumulate downstream of a normal termination codon.

3) Revise the manuscript to address as many as possible of the other major and comments or raised by the reviewers.

See responses below.

Reviewer #1:[…] The experiments are carefully designed and well-executed and the paper is well written. The results largely justify the model that the ATPase-defective UPF1 mutants somehow interfere with ribosome recycling by RLI1 following termination at the PTC, allowing the unrecycled 80S ribosomes to migrate into the 3'UTR and stall at particular sequences. While it is reasonable to propose that wild-type UPF1 would stimulate recycling by RLI1, it's unclear why UPF1 mutants defective for ATP binding (vs. hydrolysis), nor the absence of UPF1, do not give rise to scanning unrecycled 80S ribosomes that stall at the same sequences 3' of the PTC. It's also not clear whether the proposed activity of UPF1 in stimulating RLI1 recycling function is important for NMD; although it would be surprising if it were not.1) It seems feasible to test the model that ribosome recycling by ABCE1/RLI1 is impaired by the UPF1 ATPase mutants by determining whether depleting RLI1 from WT UPF1 cells, using a published degron mutant, gives rise to the same reporter decay intermediates.

We agree with this reviewer that this a critical experiment to evaluate whether ATPase-deficient UPF1 is functioning to simply preclude/prevent RLI1 activity on a prematurely terminating ribosome. We direct the reviewers again to our previously published work where we depleted RLI1 (using the inducible/repressible *GAL* promoter) and monitored accumulation of 3’ RNA fragments from our PTC-containing mRNA (Serdar et al., 2016; Supplementary Figure 4D). As reproduced there, we did detect decay intermediates from RLI1-depleted cells, however, these fragments were generally shorter and displayed significantly more heterogeneity in size compared to fragments from *UPF1-DE572AA* cells (as would be predicted based on ribosome profiling data of RLI1-depleted cells by the lab of Rachel Green at JHU; see Young et al., 2015). It is clear from our experiment that depletion of RLI1 from cells does not phenocopy the impact of ATPase-deficient UPF1 on prematurely terminating ribosomes. We discussed these previous observations in our current manuscript and used them to argue that UPF1 ATPase mutants likely work by impinging on RLI1 function (and not preventing RLI1 function; Discussion, fourth paragraph). Consistent with this interpretation, we also mention that over-expression of RLI1 from a high-copy plasmid does not reduce accumulation of (or alter the nature of) the 3’ RNA decay fragments that accrue in *UPF1-DE572AA* cells – as might be expected if mutant UPF1 was, for example, competing against RLI1 for binding to the terminating ribosome.

Incidentally, despite these results reported in 2016, we did perform PE on RLI1-depleted cells containing our *PGK^PTC125^* reporter mRNA and show these data as Author response image 2. As you will see, depletion of RLI1 did not give rise to the +17 and +20 cDNA products that we see in the presence of *UPF1-DE572AA.* In fact, we fail to see accumulation of strong cDNA products at any specific position despite clearly detecting full length (FL) cDNA, consistent with the low level and heterogenous distribution of 3’ UTR ribosomes in RLI1-depleted cells detected by ribosome profiling (Young et al., 2015). You may note that we observe a reduction in the level of the 3’ decay intermediates found in the UPF1 ATPase mutant during the RLI1 depletion time course (note it takes ~6 hrs to achieve 90% depletion of Rli1p in our cells; Serdar et al., 2016). While it is tempting to suggest that this indicates a requirement for RLI1 in the accumulation of the 3’ decay fragments – which would be completely supportive of our model – we note that in these cells, ribosomes are amassing globally on mRNA 3’ UTRs and being depleted from the cytoplasmic pool. Given that this would be predicted to reduce general translation and that 3’ decay intermediate accumulation requires translation of the PTC-containing mRNA (as specifically shown in Serdar et al., 2016), we feel that this experiment neither adds to our observation reported in 2016 (that 3’ RNA decay fragments are distinct in *UPF1*-*DE572AA* versus RLI1-depleted cells) nor provides definitive support for our model, so we prefer to exclude this data from our current submission.

**Author response image 2. respfig2:** 

2) It would enhance the story if they were able to map the 3' ends of the presumed stalled 80S ribosomes downstream of the PTC in order to confirm protection of 3'UTR mRNA sequences by entities with a ribosome-size footprint. They tried conventional ribosome profiling to no avail, but perhaps crosslinking could be attempted in an effort to stabilize the putative scanning ribosome.

We appreciate this suggestion and whole-heartedly agree that this data would provide additional *direct* evidence in support of our interpretation that a stalled 80S ribosome resides at the 5’ end of the 3’ decay intermediate. We attempted to gather this evidence in two different ways. First, as mentioned, we attempted ribosome profiling which aimed to define the precise positioning of ribosomes on the 3’ decay fragments. Despite extensive experience with this method (see for example Smith et al., 2014), we failed to map any ribosome protected fragments associated with the 3’ decay fragment – this is despite deep library coverage and strong biochemical evidence that one or more is bound to this intermediate (i.e. polyribosome analysis and ribosome IP). We think this our inability to map the ribosome footprints by this method is likely due to dissociation of the ribosomes during the extensive in vitro processing steps associated with our rigorous ribosome profiling protocol that includes recovering ribosome-protected fragments created by RNase digestion from the monosome fraction of sucrose gradients. We did not attempt ribosome profiling using cells that were subjected to crosslinking, and are not aware of any published method or report where this has been successfully performed.

Second, we attempted toeprint analysis which employs PE as a means to map the 3’ most position of an RNA-bound protein. We endeavored to detect PE products for *GFP^PTC125^* mRNA from *UPF1-DE572AA* cell lysates with or without chemical crosslinking (formaldehyde at a concentration and for a duration we have previously shown efficiently crosslinks proteins to RNA without causing extensive inter-transcript complex formation; see Smith and Baker, Methods Mol Biol 2017). Unfortunately, data was messy and the results inconclusive owing from what we believe are challenges associated with generating cDNA products from an mRNA template that is bejeweled with covalently bound RNA-binding proteins (and only a single mRNA population in a sea of transcripts in the cell extract). We note that in our assessment, toeprint analysis has been most successfully employed using nuclease-treated cell extracts and in vitro transcribed templates, which we do not confidently feel would recapitulate NMD-mediated decay. We would be delighted, however, if our reviewers can point us to any alternative approaches that we might try to get at this precise question.

Reviewer #2:[…]Essential revisions:1) Figure 4D: Based on deletion analyses which result in the disappearance of bands corresponding to decay fragments, without corresponding appearance of a band elsewhere, the authors conclude that, "upon collision with a terminating ribosome, the stalled ribosome is displaced and does not re-establish a stable interaction with the downstream RNA." I think the general applicability of this conclusion is unclear without more information about why some sequences do not support robust fragment generation. What if, for example, a reporter was constructed in which the sequence used in Figure 1—figure supplement 1D that generates a +31 fragment (with additional bands at +43 and +51) was inserted downstream of the 15 nt deletion reporter sequence? To be clear, I'm not asking for this experiment to be completed-just offering it as an illustration that alternatives to the stated conclusion are possible.

See response to this inquiry under Essential revisions (point 1) above.

2) I suggest moving Supplementary Figure 3A to the main text so all assays of protein expression in the experiment in Figure 3B can be evaluated as a whole. Likewise, incorporating the model and RNA abundance measurements from Supplementary Figure 5 to the main Figure 5 would make it easier to interpret the data currently in Figure 5. Also, measures of error and stats should be provided if possible.

We thank the reviewer for these suggestions and have moved Supplementary Figures 3A and 5B to their respective main figures (and added required text to the figure legends and Results section accordingly). We note that we have performed Northern blot analysis of *CYH2* transcripts on RNA isolated from the various UPF1 active site mutants only twice; the values provided in the figure are for the image shown, but we have added to the figure legend that this experiment has been done in duplicate and that the abundance of *CYH2* pre-mRNA (the NMD substrate) normalized to *CYH2* mRNA (NMD insensitive) differs less than 10% between experiments.

3) Figure 4A: I think the statement that "the nucleotide distance from the PTC is not a critical factor in determining how XRN1 progression is impeded" is not fully supported. Based on the overall data presented, position likely plays some role, as the products characterized are in a small range downstream of the TC and it is clear that certain positions are incompatible with cleavage or protection of the resulting fragment. It would be better to state that fragment accumulation is likely determined by a combination of sequence and position.

The cumulative data from our various PTC-containing *PGK1* mRNA reporters show that the 5’ termini of the 3’ RNA decay fragments are found at a number of distances downstream of the PTC (e.g. +17, +20, variable distances between +25 and +31, +43, +51) clearly indicating distance from the PTC alone does not determine where unrecycled ribosomes stall and block to XRN1 progression. However, we cannot rule out that there is some influence of distance (as indicated by the reviewer, some positions are incompatible [due to their proximity to the PTC] and the range of observed stall sites is restricted to the limited number of PTC-containing mRNA we analyzed); given this we have modified the text to more clearly reflect this limitation of our data.

Reviewer #3:This manuscript represents a follow-up of the work presented in Serdar et al., 2016, in which they showed that expressing of an ATPase-deficient UPF1 mutant in yeast caused the accumulation of 3´RNA decay fragments of NMD-sensitive RNAs whose size was coincident with a block of XRN1-dependent 5´-to-3´exonucleolytic degradation around the PTC position. Based on their data, they concluded that UPF1-mediated ATP hydrolysis was required for ribosome disassembly at PTCs.Here they now used primer extension assays to map the 5´end of these RNA fragments originating from a GFP^PTC125^ reporter gene and detect two bands 20 and 17 nucleotides downstream of the PTC (Figure 1). The 5´ ends of the decay fragments from GFP reporters with PTCs in two different positions also mapped downstream of the PTC, but with different distances and band patterns: GFP^PTC67^ showed bands around +20 and +12, and GFP^PTC135^ many bands at different positions between +51 and +9 (Figure 1—figure supplement 1).A control I very much would like to see here: Are such fragments also detected downstream of the normal GFP stop codon in the GFP WT reporter when UPF1 fails to hydrolyze ATP? In other words, is this phenomenon PTC-specific?Immunoprecipitation of tagged ribosomes co-precipitated mainly full-length reporter RNA but also the 3´decay fragment in cell expressing the UPF1 DE572AA mutant, suggesting that these RNA fragments are associated with ribosomes (Figure 2 and Figure 2—figure supplement 1). How do the ribosomes get to these positions downstream of the stop codon? The authors could show that this seems not to occur due to readthrough, because (i) no readthrough was detected in cells expressing the UPF1 mutant and (ii) introducing a frameshift did not change the pattern of these bands (Figure 3 and Figure 3—figure supplement 1). The mapped 5´ends of these 3´decay fragments do neither occur at a specific distance from the PTC nor depend on the identity of the stop codon but rather depend on the sequence context downstream of the PTC (Figure 4 and Figure 4—figure supplement 1). While the experiments shown in Figure 4 and Figure 4—figure supplement 1 aim at gaining some mechanistic insight into how these XRN1-resistant decay fragments might arise, they appear to stop halfway.

See our response to this particular comment in the Essential revisions (point 1) section above.

To corroborate the conclusion that the sequence context is important, the single nucleotide substitution approach (Figure 4C) should be extended to the other two PTCs and critical residues and some sort of a "consensus sequence" might be revealed by comparing the results from all three contexts.

We agree with the reviewer that determination of a *consensus sequence* for which 3’ UTR ribosomes prefers to stall could provide exciting mechanistic insight into how mutant UPF1 impinges on RLI1 and/or the prematurely terminating ribosome. We had this very idea in mind as we interrogated the sequence within 50 nts of the 5’ termini of the decay fragments that accumulate for our various PTC-containing reporter transcripts – but were unable to identify either a sequence motif or putative RNA secondary structure. While we agree that analysis of a battery of additional point mutations within this region for a number of our reporters might help to identify such a consensus, we have opted to apply a more genomic approach where libraries of transcripts with randomized sequences within the +17 and +20 regions of the *PGK1^PTC125^* mRNA reporter will be correlated with the accumulation of 3’ RNA decay fragments in vivo and measurement of 5’ ends by RNA-Seq. We think this unbiased and more comprehensive analysis of the sequence landscape around the site of the stalled ribosomes is much more likely to provide insight into the sequence requirements for ribosome stalling within the 3’ UTR. This experiment has been delayed due to COVID-19, but we feel is nevertheless beyond the scope of this current manuscript; we hope that the reviewer agrees. We have altered the text associated with Figure 4C to state that additional analyses are required to establish how sequence context influences the site of 3’ UTR ribosome stalling in our UPF1 ATPase mutants.

Similarly, deletions with increasing length as in Figure 4D should also been tested for the other two PTCs to further corroborate the hypothesis that upon collision with the terminating ribosomes positioned at the PTC, these 3´ fragments disappear.

See our response to this particular comment in the Essential revisions (point 2) section above.